# HTMFORMER: HYBRID TIME AND MULTIVARIATE TRANSFORMER FOR TIME SERIES FORECASTING

## ABSTRACT

Transformer-based methods have achieved impressive results in time series forecasting. However, existing Transformers still exhibit limitations in sequence modeling as they tend to overemphasize temporal dependencies. This incurs additional computational overhead without yielding corresponding performance gains. We find that the performance of Transformers is highly dependent on the embedding method used to learn effective representations. To address this issue, we extract multivariate features to augment the effective information captured in the embedding layer, yielding multidimensional embeddings that convey richer and more meaningful sequence representations. These representations enable Transformer-based forecasters to better understand the series. Specifically, we introduce **Hybrid Temporal and Multivariate Embeddings (HTME)**. The HTME extractor integrates a lightweight temporal feature extraction module with a carefully designed multivariate feature extraction module to provide complementary features, thereby achieving a balance between model complexity and performance. By combining HTME with the Transformer architecture, we present HTMformer, leveraging the enhanced feature extraction capability of the HTME extractor to build a lightweight forecaster. Experiments conducted on eight real-world datasets demonstrate that our approach outperforms existing baselines in both accuracy and efficiency.

## 1 INTRODUCTION

Long-term time series forecasting holds significant importance in various fields such as finance and economics (Andersen et al., 2006), climate science (Mudelsee, 2019), healthcare (Zeger et al., 2006), geophysics (Gubbins, 2004), industrial monitoring (Truong et al., 2022), *etc.* Recently, Transformer (Vaswani et al., 2017) and its variants have achieved tremendous success in time series forecasting, showing great promise. Current Transformer-based forecasters often overemphasize modeling temporal dependencies. However, the effective information within the temporal dimension is limited.

Unlike natural language sequences, time series data, particularly in domains such as transportation, meteorology, and electricity systems, are influenced by specific physics-based systems, *e.g.*, traffic flow is largely governed by the real-world road network. Therefore, multivariate correlations are essential for time series forecasting

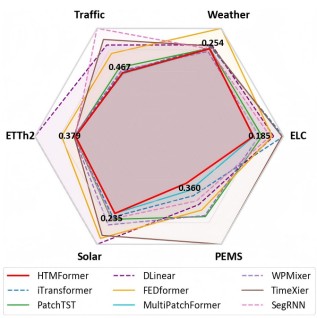

Figure 1: Performance (MSE) of HTMformer. Smaller image area yields better performance.

tasks (Zheng & Sun, 2024). Two key observations motivate our work: **1)** the vast majority of features in time series are contained within the temporal dimension, and the depth of exploration in this dimension significantly influences prediction accuracy (Hyndman & Athanasopoulos, 2021); **2)** the multivariate dimension also contains a large number of features (Pravilovic et al., 2013), and fully exploiting multivariate features can further boost performance.

Consequently, existing models incur increasing computational overhead without significant accuracy improvements. Some predictors take multivariate correlations into account, focusing on explicitly modeling dependencies across variables. However, the primary characteristics of time series data reside in the temporal dimension. Such an approach incurs substantial computational overhead while interfering with the extraction of temporal features as they share a common latent space or backbone

Figure 2: Previous works have focused on redesigning Transformer architectures or attention mechanisms. We propose a novel embedding strategy.

network. Existing methods have yet to achieve a optimal balance between feature extraction across both dimensions and computational efficiency.

The performance of Transformer-based forecasters is highly dependent on the design of the embedding layer. Before applying multi-head self-attention to the embeddings, effective feature extraction is critical to enhancing the model's sequence modeling capacity (Li et al., 2023a). We compared the ground-truth values and the predictions obtained under different embedding strategies on four datasets (see Appendix A). Specifically, models such as iTransformer (Liu et al., 2024b) and PatchTST (Nie et al., 2023), which employ an inverted embedding strategy or patch-based embedding strategy for deeper time-series representation learning, achieve significantly better performance than the vanilla Transformer, which only uses simple convolutional embeddings. Meanwhile, we also observed that as the input length increased, the prediction loss failed to markedly decrease and even tended to rise. These suggest that the modeling capacity of Transformer-based forecasters is inherently constrained by the quality of embeddings, and this limitation cannot be eliminated by increasing the lookback length.

To mitigate this issue, this paper proposes to jointly capture temporal and multivariate features in the embedding layer, thereby yielding semantically rich embedding representations to overcome the inherent limitations of the conventional embedding strategy (see Appendix B). Leveraging this design, we construct a novel Transformer-based forecaster, as illustrated in Figure 2. The proposed forecaster achieves state-of-the-art performance on real-world benchmarks, as shown in Figure 1. The contributions of this work are as follows:

- We propose the Hybrid Temporal and Multivariate Embedding (HTME) extractor, which comprises a novel design temporal feature extractor and a lightweight multivariate feature extractor. HTME can be seamlessly integrated with any forecasting architecture. Moreover, its two parallel modules can be decoupled and individually paired with other embedding schemes. Overall, HTME is highly extensible.

- We selected eight widely-used forecasting models and developed their HTME-augmented versions. Extensive experiments demonstrate that HTME consistently enhances the ability of diverse architectures to model complex time series, particularly for Transformer framework.

- We introduce a novel forecasting model, HTMformer, as the representative HTMPredictors, to further validate the effectiveness of the HTME strategy. In this framework, the HTME serves as input to a Transformer framework. Notably, HTMformer consistently achieves state-of-the-art performance across various benchmarks.

## 2 RELATED WORK

**Transformer-Based Time Series Forecasting.** Well-designed Transformer-based forecasters have achieved substantial success in time series forecasting tasks. They are not limited by the inability of CNN-based methods (Liu et al., 2022) to capture long-term temporal dependencies (Sutskever et al., 2014; Karim et al., 2017), nor do they encounter the vanishing and exploding gradient problems (Bengio et al., 1994; Hochreiter & Schmidhuber, 2019) commonly observed in RNN-based methods (Cho et al., 2014; Lai et al., 2018; Salinas et al., 2020). In recent years, advanced models such as Informer (Zhou et al., 2021), FEDformer (Zhou et al., 2022), and MultiPatchFormer (Naghashi et al., 2025) have primarily introduced novel attention mechanisms and architectural modifications for explicitly modeling long-term dependencies and temporal interactions in time series data. Unlike prior works, we propose a novel embedding strategy for comprehensive feature extraction on the raw

input before it is fed into the attention module, thereby substantially improving the modeling capacity of diverse attention mechanisms.

**Temporal Feature Extraction.** Previous Transformer-based forecasters, such as FEDformer (Zhou et al., 2022) and MultiPatchFormer (Naghashi et al., 2025), tend to overprioritize temporal feature extraction, introducing various complex mechanisms to model temporal dependencies. This design often incurs substantial computational overhead without delivering notable performance gains. We argue that the informational content in the temporal dimension is inherently limited and should not be excessively emphasized. In contrast, HTMformer performs full temporal feature extraction solely within the embedding layer, thereby significantly reducing computational complexity.

**Multivariate Correlation.** iTransformer retains all native Transformer components without modification. Instead, it adopts an inverted input representation, enabling the self-attention model to better capture multivariate correlations and thereby yielding satisfactory performance. This underscores the importance of modeling multivariate correlations in capturing the semantics of time series. In CNN-based predictors, graph convolutional networks (Kipf & Welling, 2017; Yu et al., 2018; Guo et al., 2019) explicitly model correlations among variables by leveraging graph adjacency matrices. This paradigm unifies the temporal dynamics of the data with the multivariate structure of variables within a graph-based model, thereby substantially improving performance in time series tasks. However, such explicit modeling entails significant computational overhead. HTMformer is designed to efficiently extract multivariate features and integrate them with temporal information within the embedding layer, yielding semantically enriched embeddings to enhance the efficiency and performance of Transformers and reduce the computational costs.

## 3 HTMFORMER

This paper focuses on multivariate time series forecasting tasks, which can be formally defined as follows: Given a historical observation matrix $\mathbf{X} \in \mathbb{R}^{L \times C}$, where $L$ denotes the input sequence length and $C$ represents the number of variables, the goal is to predict the future values $\mathbf{Y} \in \mathbb{R}^{H \times C}$, with $H$ indicating the prediction horizon.

### 3.1 HTME EXTRACTOR

HTME avoids the use of complex mechanisms for modeling multivariate correlations. Instead, it employs a lightweight method to extract multivariate features, projecting them into the latent space of the temporal dimension during the embedding phase. HTME deeply extracts hybrid temporal-multivariate features from the time series.

To fully capture informative representations from the raw time series, the HTME extractor incorporates two core components: a temporal feature extractor and a multivariate feature extractor, which enable the independent modeling of two dimensions. This parallel design reduces interference between features from different dimensions. Finally, we employ a weighted summation for feature fusion, enabling multivariate features to complement temporal features, thereby yielding a richer representation. The HTME extractor takes raw multivariate time series as input and outputs embedding representations that fuse informative features, as illustrated in Figure 3.

Within the two branches, we adopting a bottom-up fusion followed by a top-down decomposition strategy. Bottom-up fusion encourages the model to focus on a single dimension at a time, minimizing the interference from other dimensions. Top-down decomposition progressively extracts features across multiple scales, while guiding the model to emphasize short-term patterns, resulting in more effective feature embeddings.

### 3.1.1 TEMPORAL FEATURE EXTRACTOR

Appendix A clearly shows that time series exhibit both short-term and long-term correlations. It is worth noting that short-term correlations are predominant, while the influence of long-term correlations should not be neglected. The strategy of first merging and then progressively decomposing features effectively guides the model to focus step-by-step on the most informative patterns.

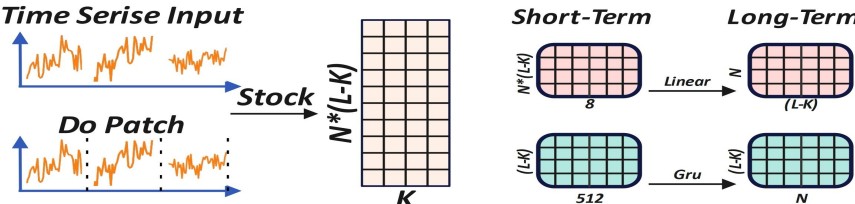

Figure 3: HTME consists of two independent and parallel feature extraction modules. It employs a strategy of first merging the outputs and then hierarchically decomposing the resulting representation.

First, we perform hierarchical merging of the raw time series. Specifically, we adopt a patching strategy to segment short-term temporal representations from the original sequence, and iteratively merge all channels to obtain a short-step representation $T^{N \times (L-K), K}$ .It is worth noting that the process merely partitions the time series without introducing trainable parameters at this stage.

$$P_1, P_2, P_3, \ldots, P_{L-K} = X_{1:1+K}, X_{2+K}, X_{3+K}, \ldots, X_{L-K:L} \tag{1}$$

$$T^{N \times (L-K), K} = \text{Stack}(P_1, P_2, P_3, \ldots, P_{L-K}) \tag{2}$$

This module first prioritizes the deep extraction of short-term temporal features, which are particularly important. Specifically, we apply multiple convolutional filters to capture short-term temporal patterns at multiple scales. We then fuse the convolutional outputs via a linear layer to obtain a feature representation enriched with short-term dependencies.

$$E^{N \times (L-K), 8} = \text{Linear}(\text{Conv}(T^{N \times (L-K), K})) \tag{3}$$

Finally, we unfold the temporal dimension and employ a second linear layer to capture the long-term dependencies within the sequence, yielding a deep embedding that encapsulates both short-term and long-term temporal dependencies.

$$D_{\text{out}} = \text{Linear}(\text{Flatten}(E^{N \times (L-K), 8})) \tag{4}$$

Compared with existing embedding strategies, our method achieves a more fine-grained modeling of temporal contextual dependencies using only one convolutional layer and two linear layers.

### 3.1.2 MULTIVARIATE FEATURE EXTRACTOR

We adopt the same merge-decompose strategy as described above. It is worth noting that correlations across variables may exhibit temporal lag. A change occurring at one node may exert influence on other nodes at the subsequent time step. We segment and stack the time series to obtain $T^{N \times (L-K), K}$, following exactly the same procedure as in the aforementioned module.

Subsequently, we merge the short-term temporal dimension with the multivariate dimension, and employ a linear layer to process the multi-step variables, thereby obtaining multivariate features $N^{(L-K), 512}$ that account for short-term temporal lags.

$$N^{(L-K), 512} = \text{Linear}(T^{N \times (L-K), K}) \tag{5}$$

Similar to Eq. (4), we employ multiple convolutional filters to extract multivariate features at different scales, resulting in a more fine-grained multi-scale multivariate feature block. As this module is substantial in size, we therefore apply a specialized GRU network to aggregate the multi-scale features and obtain the integrated representation $B^N$.

$$B^N = \text{Gru}(\text{Conv}(N^{(L-K), 512})) \tag{6}$$

Considering that the correlations among multivariate variables may differ across time stamps, we apply L convolutional kernels to project the integrated representation $B^N$ onto the temporal dimension, thereby obtaining the final representation.

$$V_{\text{out}} = \text{Conv}(B^N) \tag{7}$$

Figure 4: Architecture of HTMformer. HTME directly replaces the original iTransformer embeddings. It also employs inverted input to enable the attention mechanism to capture channels correlations.

here, $D_{\text{out}}, V_{\text{out}} \in \mathbb{R}^{N \times D}$. We define a learnable fusion weight $\alpha$ to adaptively balance the contributions of the two modules to the downstream model, thereby enhancing the scalability of HTME for adaptation to datasets with diverse characteristics. The final output is the sum of the two modules:

$$Y_{\text{in}} = \alpha D_{\text{out}} + (1 - \alpha) V_{\text{out}}, \tag{8}$$

where $Y_{\text{in}} \in \mathbb{R}^{N \times D}$ denotes the embedded representation that fuses temporal and multivariate features. HTME is fed into a vanilla Transformer encoder to generate the predictive representations.

## 3.2 STRUCTURE OVERVIEW

We construct HTMformer based on iTransformer, a state-of-the-art and generic Transformer architecture. It keeps the native components of the original Transformer intact. This implies that other components within HTMformer can be flexibly interchanged with their respective variants. As shown in Figure 4, the model includes an HTME extractor, a vanilla Transformer encoder layer, and a projection layer. RevIN (Kim et al., 2022) is a normalization technique, which can help models learn and generalize better. Each data batch is first normalized with RevIN before being fed into the model.

In HTMformer, shown in Figure 4 (**Middle**), the overall formulation for predicting the future sequence $Y \in \mathbb{R}^{H \times C}$ from the historical sequence $X \in \mathbb{R}^{L \times C}$ is as follows:

$$\begin{aligned} X_{\text{in}} &= \text{CAT}(X, T), & Y_{\text{in}} &= \text{HTMEE}(X_{\text{in}}), \\ Y_{\text{out}} &= \text{Encode}(Y_{\text{in}}), & Y &= \text{Project}(Y_{\text{out}}). \end{aligned} \tag{9}$$

Following RevIN normalization, timestamps $T$ are appended as an additional variable to the sequence $X \in \mathbb{R}^{L \times C}$ before being fed into the model. $X_{\text{in}} \in \mathbb{R}^{L \times N}$, where $N$ denotes the multivariate dimension augmented with the timestamp. $Y_{\text{in}} \in \mathbb{R}^{N \times D}$ denotes the hybrid temporal and multivariate embedding representations after the HTME extractor, where the embedding dimension of each variable is $D$. $Y_{\text{out}} \in \mathbb{R}^{N \times D}$ represents the predictive features produced by the Transformer encoder. Finally, a linear model serves as the projection layer to generate the final output $Y \in \mathbb{R}^{H \times C}$.

## 3.3 ENCODER AND PROJECTION

The Transformer encoder layer is composed of a self-attention module and a position-wise feed-forward network. We adopt an inverted input design, which enables the self-attention mechanism to directly model global channel correlations. This inverted structure avoids forced alignment of variables within timestamps and facilitates the concatenation of temporal and multivariate dimensions. Consequently, it enables the attention mechanism to learn more appropriate sequence representations.

The attention mechanism projects $Y_{\text{in}} \in \mathbb{R}^{N \times D}$ to generate query matrix $Q_k$, key matrix $K_k$, and value matrix $V_k$, where $d_k$ denotes the projection dimension for each attention head. Then, the scaled

dot-product is used to derive the attention output $O_k \in \mathbb{R}^{N \times d_k}$:

$$(O_k)^T = \text{Softmax}\left(\frac{Q_k K_k^T}{\sqrt{d_k}}\right) V_k. \tag{10}$$

The self-attention block additionally consists of layer normalization and a position-wise feed-forward network. Finally, a linear projection head serves as the decoder, mapping the learned representations to the target dimensionality and yielding the final predictions. iTransformer (Liu et al., 2024b) observes that this Transformer-based predictor architecture is more efficient for time series forecasting tasks.

### 3.4 COMPLEXITY ANALYSIS

HTME consists of two parallel extraction modules. The time and space complexities of the Feature Extraction Modules are both O(LN). For each dimension, the computational complexity is linear, which guarantees the high efficiency and scalability of HTME.Experiments show that for most models, HTME only adds a slight increase to their computational overhead.

HTMformer's computational complexity is primarily dictated by the Transformer encoder, as the introduction of the inverted input design results in both time and memory complexity scaling as $\mathcal{O}(N^2)$. Compared with the state-of-the-art Transformer forecasters, HTMformer attains lower complexity because the number of variate tokens is fixed and independent of the lookback length, which confers clear advantages for long-term time series forecasting. Moreover, as the key component, the HTME extraction module is flexibly compatible with Transformer variants integrating efficient attention mechanisms. It allows for further reductions in computational and memory costs and broadens the model's applicability to a wider range of forecasting scenarios (see Appendix C).

## 4 EXPERIMENTS

To validate the effectiveness and robust transferability of the HTME strategy, we integrate it into a diverse set of forecasting models and assess its impact on their performance. Moreover, we conduct a comprehensive evaluation of the proposed HTMformer on diverse time series forecasting and analysis tasks to further validate the advantages of the HTME strategy. The detailed experimental configurations and implementation details can be found in Appendix D.

### 4.1 EXPERIMENT SETTINGS

All models are evaluated under identical settings, following the Time-Series-Library (Wang et al., 2024b), to ensure the fairness of comparison and the reliability of the drawn conclusions.

**Datasets.** We evaluated the performance of HTMformer on eight widely used datasets, including Weather, Traffic, Electricity, ETTh2 adopted in Autoformer (Wu et al., 2021), Solar-Energy proposed in LSTNet (Lai et al., 2018), and three PEMS datasets (PEMS03, PEMS04, PEMS08) investigated in SCINet (Liu et al., 2022). These datasets span diverse domains and sampling frequencies, providing a comprehensive experimental scenario for model evaluation.

**Baselines and metrics.** We employ eight carefully selected state-of-the-art Transformer-based models and widely recognized non-Transformer-based models in the time series forecasting domain as baselines, encompassing iTransformer (Liu et al., 2024b), PatchTST (Nie et al., 2023), FEDformer (Zhou et al., 2022), Dlinear (Zeng et al., 2023), WPMixer (Murad et al., 2025), Multi-PatchFormer (Naghashi et al., 2025), TimeMixer (Wang et al., 2024a), and SegRNN (Lin et al., 2023). Mean Squared Error (MSE) and Mean Absolute Error (MAE) are utilized as the assessment criteria.

### 4.2 PERFORMANCE PROMOTION WITH HTME

To assess the effectiveness and scalability of HTME, Transformer Attention (Vaswani et al., 2017) and its variants are integrtrated into the HTMformer framework, as shown in Table 1. Specifically, HTMformers achieve consistent improvements across different variants, with average performance gains of *35.8%*, *34.3%*, *43.6%*, and *31.9%* for Transformer, Reformer (Kitaev et al., 2020), Informer (Zhou et al., 2021), Flowformer (Ma et al., 2023), and Flashformer (Dao et al., 2022),

Table 1: Performance improvementst by HTME. Full results can be found in Appendix E.1.

| Models | | Transformer | | Reformer | | Informer | | Flowformer | | Flashformer | |
|---|---|---|---|---|---|---|---|---|---|---|---|
| Metric | | MSE | MAE | MSE | MAE | MSE | MAE | MSE | MAE | MSE | MAE |
| Electricity | Original | 0.271 | 0.343 | 0.342 | 0.417 | 0.379 | 0.442 | 0.272 | 0.370 | 0.268 | 0.364 |
| | +HTME | 0.185 | 0.272 | 0.201 | 0.292 | 0.202 | 0.292 | 0.198 | 0.289 | 0.194 | 0.287 |
| | Promotion | 35.3% | 20.6% | 41.2% | 29.9% | 46.7% | 33.9% | 27.2% | 19.4% | 27.6% | 21.1% |
| Weather | Original | 0.651 | 0.572 | 0.454 | 0.458 | 0.644 | 0.557 | 0.631 | 0.559 | 0.633 | 0.559 |
| | +HTME | 0.254 | 0.278 | 0.257 | 0.276 | 0.254 | 0.280 | 0.254 | 0.279 | 0.253 | 0.280 |
| | Promotion | 60.8% | 51.3% | 45.1% | 39.9% | 60.5% | 49.7% | 57.9% | 50.0% | 60.0% | 49.9% |
| Traffic | Original | 0.666 | 0.366 | 0.706 | 0.389 | 0.825 | 0.465 | 0.655 | 0.359 | 0.661 | 0.362 |
| | +HTME | 0.467 | 0.312 | 0.475 | 0.314 | 0.488 | 0.323 | 0.481 | 0.321 | 0.474 | 0.319 |
| | Promotion | 29.8% | 17.3% | 32.7% | 19.2% | 40.8% | 30.5% | 26.5% | 10.5% | 28.2% | 11.8% |
| Promotion Avg | | 35.8% | | 34.6% | | 43.6% | | 31.9% | | 33.1% | |

respectively. These results demonstrate the HTME framework significantly strengthens the modeling capacity of Transformer-based forecasters, allowing HTMformer frames for flexible utilization of various attention mechanisms tailored to different application scenarios.

To demonstrate the strong scalability of HTME strategy, we apply it to predictors with diverse architectures, including the RNN-based SegRNN, and the linear-based DLinear. The results show that HTME consistently improves the forecasting performance across all these models.For these two models, we only supplement the original input with multivariate features.

More importantly, the temporal feature extraction module and the multivariate feature extraction module within the HTME layer operate as entirely decoupled components. Consequently, the multivariate representations produced by HTME can be seamlessly integrated with other embedding schemes that emphasize temporal characteristics, thereby further enhancing the applicability and versatility of HTME.

Table 2: The HTM version augments the original sequence with multivariate features, the -A version is an ablation variant operating solely on multivariate features. The best results are marked in red.

| Models | **HTMDLinear** | | DLinear | | DLinear-A | | **HTMSegRnn** | | SegRnn | | SegRnn-A | |
|---|---|---|---|---|---|---|---|---|---|---|---|---|
| Metric | MSE | MAE | MSE | MAE | MSE | MAE | MSE | MAE | MSE | MAE | MSE | MAE |
| Electricity | **0.219** | **0.314** | 0.225 | 0.318 | 0.500 | 0.531 | **0.190** | **0.283** | 0.194 | 0.288 | 0.416 | 0.470 |
| Weather | **0.261** | **0.286** | 0.269 | 0.289 | 0.274 | 0.348 | **0.249** | **0.296** | 0.254 | 0.301 | 0.258 | 0.308 |
| Traffic | **0.609** | **0.383** | 0.672 | 0.418 | 0.976 | 0.546 | **0.750** | **0.392** | 0.761 | 0.397 | 1.263 | 0.590 |

### 4.3 LONG-TERM FORECASTING

Long-term forecasting holds significant importance in various application domains, including meteorology, traffic management, and energy utilization. We conduct a comprehensive evaluation of the effectiveness of HTMformer in long-term forecasting. Table 3 presents the experimental forecasting results. Within each row, the lowest averaged MSE and MAE values across four prediction horizons are highlighted in **red**, and the second-lowest values are underscored in blue. HTMformer achieves the best average performance across the majority of datasets, especially on high-dimensional datasets such as Traffic. We further provide a visualization of the long-term forecasting results produced by HTMformer (see Appendix I). HTMformer effectively captures the periodic characteristics of time series, delivers accurate trend forecasts for complex sequences, and demonstrates strong robustness against large-scale fluctuations. In conclusion, HTMformer demonstrates significant performance improvements and tangible practical benefits in long-term forecasting tasks.

Table 3: Average performance for long-term forecasting over prediction horizons $H \in \{96, 192, 336, 720\}$ with fixed lookback $L = 96$. Notably, MPFormer is short for MultiPatchFormer. Full results are listed in Appendix E.2.

| Models | HTMformer (Ours) | | iTransformer (2024b) | | PatchTST 2023 | | DLinear 2023 | | FEDformer 2022 | | MPFormer 2025 | | WPMixer 2025 | | TimeMixer 2024a | | SegRNN 2023 | |
|---|---|---|---|---|---|---|---|---|---|---|---|---|---|---|---|---|---|---|
| Metric | MSE | MAE | MSE | MAE | MSE | MAE | MSE | MAE | MSE | MAE | MSE | MAE | MSE | MAE | MSE | MAE | MSE | MAE |
| ECL | **0.185** | **0.272** | 0.244 | 0.290 | 0.202 | 0.286 | 0.225 | 0.318 | 0.222 | 0.333 | 0.186 | 0.273 | 0.196 | 0.281 | 0.241 | 0.328 | 0.218 | 0.302 |
| Weather | 0.254 | 0.277 | 0.259 | 0.279 | 0.255 | 0.278 | 0.265 | 0.316 | 0.313 | 0.362 | 0.252 | 0.274 | **0.246** | **0.273** | 0.262 | 0.286 | 0.253 | 0.3 |
| Traffic | 0.467 | 0.312 | 0.474 | 0.317 | 0.507 | 0.324 | 0.672 | 0.418 | 0.610 | 0.378 | **0.462** | **0.304** | 0.484 | 0.338 | 0.713 | 0.445 | 0.795 | 0.408 |
| ETTh2 | **0.379** | **0.399** | 0.383 | 0.406 | 0.382 | 0.404 | 0.563 | 0.519 | 0.442 | 0.454 | 0.381 | 0.406 | 0.387 | 0.410 | 0.385 | 0.409 | 0.381 | 0.414 |
| Solar | **0.235** | **0.265** | 0.246 | 0.278 | 0.253 | 0.289 | 0.329 | 0.400 | 0.311 | 0.392 | **0.235** | 0.267 | 0.27 | 0.303 | 0.302 | 0.323 | 0.252 | 0.304 |
| PEMS03 | **0.289** | **0.369** | 0.360 | 0.421 | 0.502 | 0.51 | 0.442 | 0.498 | 0.467 | 0.503 | 0.336 | 0.413 | 0.511 | 0.497 | 0.726 | 0.614 | 0.413 | 0.456 |
| PEMS04 | **0.284** | **0.376** | 0.406 | 0.455 | 0.624 | 0.575 | 0.441 | 0.494 | 0.471 | 0.507 | 0.381 | 0.441 | 0.586 | 0.548 | 0.800 | 0.652 | 0.452 | 0.485 |
| PEMS08 | **0.508** | **0.448** | 0.598 | 0.498 | 0.719 | 0.573 | 0.69 | 0.556 | 0.758 | 0.61 | 0.521 | 0.462 | 0.724 | 0.557 | 0.956 | 0.664 | 0.621 | 0.511 |
| Count | **6** | **6** | 0 | 0 | 0 | 0 | 0 | 0 | 0 | 0 | 3 | 1 | 1 | 1 | 0 | 0 | 0 | 0 |

## 4.4 SHORT-TERM FORECASTING

Short-term time series forecasting tasks are also prevalent across a wide range of application domains. Table 4 presents the experimental results on the PEMS (PEMS03, PEMS04, and PEMS08) datasets. The proposed HTMformer demonstrates better performance across all three PEMS datasets. It outperforms the second-best model, MultiPatchFormer, yielding a *21.7%* reduction in MSE and a *12.0%* reduction in MAE, respectively. We attribute this performance gain to the exploitation of rich multivariate representations in HTME. Since short-term forecasting, constrained by the limited prediction length, does not entail intricate long temporal dependencies, it renders inter-variable correlations more salient. These results validate the robust performance of the HTMformer on short-term forecasting tasks, further complementing its strong performance in long-term forecasting.

Table 4: Short-term forecasting results, averaged across four prediction lengths $H \in \{12, 24, 48, 96\}$ with a fixed lookback window of $L = 96$. Full results are available in Appendix E.3.

| Models | HTMformer (Ours) | | iTransformer (2024b) | | PatchTST 2023 | | DLinear 2023 | | FEDformer 2022 | | MPFormer 2025 | | WPMixer 2025 | | TimeMixer 2024a | |
|---|---|---|---|---|---|---|---|---|---|---|---|---|---|---|---|---|
| Metric | MSE | MAE | MSE | MAE | MSE | MAE | MSE | MAE | MSE | MAE | MSE | MAE | MSE | MAE | MSE | MAE |
| PEMS03 | **0.138** | **0.245** | 0.181 | 0.283 | 0.267 | 0.350 | 0.278 | 0.377 | 0.206 | 0.325 | 0.172 | 0.278 | 0.262 | 0.336 | 0.348 | 0.391 |
| PEMS04 | **0.138** | **0.249** | 0.216 | 0.308 | 0.323 | 0.385 | 0.295 | 0.388 | 0.206 | 0.328 | 0.207 | 0.305 | 0.304 | 0.369 | 0.387 | 0.426 |
| PEMS08 | **0.187** | **0.278** | 0.238 | 0.31 | 0.297 | 0.361 | 0.377 | 0.414 | 0.286 | 0.359 | 0.213 | 0.296 | 0.314 | 0.360 | 0.392 | 0.408 |
| Count | **3** | **3** | 0 | 0 | 0 | 0 | 0 | 0 | 0 | 0 | 0 | 0 | 0 | 0 | 0 | 0 |

Compared with iTransformer (Liu et al., 2024b), which explicitly models multivariate correlations, and PatchTST (Nie et al., 2023), which excels at capturing fine-grained temporal patterns, HTMformer achieves broadly superior performance on both long-term and short-term forecasting tasks. Even when compared with the previously best-performing Transformer-based forecaster, Multi-PatchFormer, HTMformer consistently achieves either superior or comparable predictive accuracy. This performance gain is particularly pronounced on high-dimensional time series datasets, such as Traffic and Solar-Energy. Such effectiveness is attributed to the use of the HTME extractor, which facilitates multi-dimensional feature fusion in time series. HTME adaptively integrates temporal and multivariate representations to achieve comprehensive, multidimensional feature complementarity. Furthermore, it encapsulates richer and more discriminative semantic representations, enabling Transformer-based forecasters to more fully leverage sequence information. This demonstrates the broad applicability across diverse datasets, thereby highlighting the effectiveness of HTME.

Table 5: Results (Average) of different variants under prediction lengths $H \in \{96, 192, 336, 720\}$. The input lookback $L$ is set to 96.

| Models | HTMformer | | HTMformerV1 | | HTMformerV2 | | iTransformer | | iTransV3 | | MPFormer | |
|--------|-----------|-----|-------------|------|-------------|------|--------------|------|----------|------|----------|------|
| Metric | MSE | MAE | MSE | MAE | MSE | MAE | MSE | MAE | MSE | MAE | MSE | MAE |
| ECL | 0.185 | 0.272 | 0.184 | 0.273 | 0.275 | 0.357 | 0.244 | 0.290 | 0.194 | 0.288 | 0.186 | 0.273 |
| Weather | 0.254 | 0.277 | 0.253 | 0.278 | 0.262 | 0.290 | 0.259 | 0.279 | 0.252 | 0.267 | 0.252 | 0.274 |
| Traffic | 0.467 | 0.312 | 0.477 | 0.318 | 0.748 | 0.440 | 0.474 | 0.317 | 0.473 | 0.321 | 0.462 | 0.304 |
| ETTh2 | 0.379 | 0.399 | 0.381 | 0.405 | 0.444 | 0.444 | 0.383 | 0.406 | 0.384 | 0.406 | 0.381 | 0.406 |
| Solar | 0.235 | 0.265 | 0.243 | 0.278 | 0.288 | 0.311 | 0.246 | 0.278 | 0.244 | 0.280 | 0.235 | 0.267 |
| PEMS03 | 0.289 | 0.369 | 0.340 | 0.410 | 0.351 | 0.403 | 0.360 | 0.421 | 0.294 | 0.375 | 0.336 | 0.413 |
| PEMS04 | 0.284 | 0.376 | 0.391 | 0.431 | 0.315 | 0.402 | 0.406 | 0.455 | 0.300 | 0.386 | 0.381 | 0.441 |
| PEMS08 | 0.508 | 0.448 | 0.594 | 0.492 | 0.558 | 0.465 | 0.598 | 0.498 | 0.485 | 0.434 | 0.521 | 0.462 |

## 5 MODEL ANALYSES

### 5.1 ABLATION STUDIES

To evaluate the contribution and reliability of each module within HTMformer, we conducted ablation studies (see Appendix F). Specifically, we examined two different variants of HTMformer: 1) **HTMformerV1**, which utilizes only the temporal feature extraction module of the HTME extractor; 2) **HTMformerV2**, which utilizes only the multivariate feature extraction module of the HTME extractor. We also examined another variant 3) **iTransformerV3 (iTransV3)**, which integrates the multivariate feature extraction module of the HTME extractor into the embedding layer of iTransformer. We adopt iTransformer (Liu et al., 2024b) as the state-of-the-art baseline. Sharing a highly similar architectural design with HTMformer, both models employ a dedicated embedding module and process inverted inputs before feeding them into the Transformer encoder. However, in iTransformer, the embedding layer conducts feature extraction exclusively along the temporal dimension. Notably, since all experiments were performed under identical configurations, the HTMformer variants are directly comparable to the other models. We also adopt MultiPatchFormer (Naghashi et al., 2025), attaining the second-best performance in various forecasting tasks, as a representative proxy for other forecasting models. Specific results are documented in Table 5.

**Effect of the temporal extractor.** Compared with other time series forecasting models, HTM-formerV1 already outperforms several baselines and suffers only a marginal accuracy drop even when benchmarked against state-of-the-art approaches. Blue numbers indicate the cases where HTM-formerV1 outperforms iTransformer. This observation suggests that our temporal feature extraction module has effectively captured informative temporal patterns.

**Effect of the multivariate features.** To facilitate an intuitive comparison, we evaluate HTM-formerV1 against HTMformerV2, with numerical values underlined when HTMformerV2 achieves better performance over HTMformerV1. Across most datasets, particularly those with a large number of variables, HTMformerV2 performs substantially worse than other time series forecasting models, highlighting that effective forecasting primarily depends on accurate temporal dependency modeling. Nevertheless, on certain datasets, HTMformerV2 attains superior or comparable results relative to most baseline models. This emphasizes that multivariate features are crucial and indispensable sources of information in time series forecasting.

**Effect of the hybrid strategy.** We further conduct comparative evaluations between HTMformer and HTMformerV1, as well as between iTransformerV1 and the original iTransformer. Red numbers indicate cases where the versions augmented with multivariate integration in the embedding layer outperform their original counterparts. Across most datasets, both HTMformer and iTransformerV1 consistently achieve superior performance. This provides strong evidence for the effectiveness of our proposed strategy for incorporating multivariate features into embedding representations.

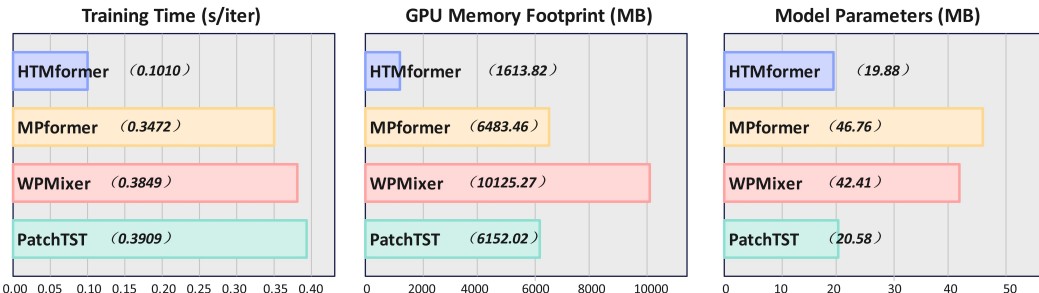

Figure 5: We compare HTMformer with three state-of-the-art models on eight benchmark datasets. The input sequence length is uniformly set to 96, and the prediction sequence length is 192.

## 5.2 MODEL EFFICIENCY

To assess the computational efficiency of HTMformer, we perform a comparative analysis of its training time, GPU memory footprint, and parameter count against state-of-the-art models, including MultiPatchFormer (Naghashi et al., 2025), WPMixer (Murad et al., 2025), and PatchTST (Nie et al., 2023). All models are evaluated on the same GPU under identical lookback windows, prediction horizons, and batch sizes to ensure fair comparison. As shown in Figure 5, using the state-of-the-art MultiPatchFormer as the reference baseline, HTMformer achieves a training runtime approximately one-third of that of MultiPatchFormer, while requiring only 20% to 45% of its GPU memory footprint. Moreover, HTMformer's model parameters are just half those of MultiPatchFormer. Importantly, these gains are achieved without sacrificing predictive accuracy, as HTMformer consistently attains superior or comparable MSE and MAE across eight benchmark datasets. These results indicate that HTMformer enables faster training and inference under the same hardware conditions, and its smaller memory footprint also enhances the model's applicability (see Appendix G).

It is noteworthy that the ablation studies and the long-term forecasting experiments were conducted under identical configurations, rendering the results directly comparable. We observe that for time series data, using patching and convolutions for local patterns and linear layers for long-range dependencies is generally sufficient for effective temporal modeling. Excessive modeling of temporal dependencies yields limited gains in forecasting accuracy while substantially increasing compute and memory overhead. In this study, we adopt a hybrid design in the embedding layer, employing two lightweight models to extract multidimensional features. The resulting hybrid embeddings extract richer informative representations from the sequences, thereby assisting Transformer-based forecasters in modeling time series. This approach not only improves predictive performance but also markedly reduces both computational demand and memory footprint.

## 6 CONCLUSION AND LIMITATIONS

Motivated by the inherent properties of multivariate time series, we introduce the HTME strategy. The HTME module jointly encodes temporal dynamics and inter-variate dependencies present in the raw sequences, yielding embedding representations that are richer and more informative than prior designs. The HTME strategy demonstrated superior performance across various experiments.

The HTME strategy can be seamlessly and efficiently integrated with various forecasting frameworks, and it consistently enhances predictive performance across diverse architectures, particularly for Transformer-based predictors. HTMformer delivers state-of-the-art forecasting performance with faster training and a reduced GPU memory footprint, thereby validating the effectiveness of injecting multivariate information into the embedding space.

Through numerous experiments, this paper further highlights that incorporating both temporal and multivariate information enables Transformer-based predictors to make more accurate predictions. The detailed discussion is provided in Appendix J. However, time series data exhibit complex correlations, and simply adding multivariate features to the temporal dimension through the embedding layer cannot fully model such a complex pattern. Jointly modeling spatiotemporal dependencies in time series remains an open problem that requires further exploration. Future improvements in Transformers can focus on devising methods to efficiently capture multivariate correlations.

## 7 AI USAGE STATEMENT

AI tools were used only for minor language polishing to standardize expressions. No AI assistance was employed for idea generation, data analysis, experiment design, or content creation. All intellectual contributions, results, and conclusions are the authors' own.

## 8 REPRODUCIBILITY STATEMENT

All relevant implementation details, such as dataset specifications, main model hyperparameters, and experimental setups, are provided in the Appendix. The source code will be released publicly once the manuscript is accepted.

## 9 ETHICS STATEMENT

This study only focuses on the domain of time series forecasting. All datasets employed are widely recognized, real-world, and publicly available. Therefore, there is no potential ethical risk.

## 10 ACKNOWLEDGEMENTS

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

The appendix is divided into several sections, each giving extra information and details.

## A  EXPERIMENTS OF DIFFERENT EMBEDDING METHODS

To demonstrate that Transformer-based forecasters have limited ability to understand sequences and the necessity of deeply mining sequence features in the embedding layer, we compared the prediction performance of Transformer (Vaswani et al., 2017), iTransformer (Liu et al., 2024b), and PatchTST (Nie et al., 2023) with different embedding approaches on four datasets: ECL, Weather, ETTh2, and Solar. The input sequence length is set to 48, 96, 144, 192, 240, and 288, with the prediction length being 96. The experimental results are shown in Figure 6.

Experimental results reveal that increasing the input sequence length initially leads to a significant decline in prediction loss. However, as the sequence length continues to grow, this benefit markedly lessens, and in some cases, prediction loss may even increase. These findings suggest that Transformer-based models exhibit limitations in proficiency in capturing effective features from raw series data. Simply increasing the lookback window does not consistently capture more informative patterns and will also incur additional computational overhead. This is because time series forecasting focuses on capturing short-term dependencies (Kim et al., 2025) and the temporal dimension inherently offers limited contextual information. Therefore, traditional Transformer-based forecasters encounter bottlenecks in reliably improving prediction accuracy. Notably, employing more sophisticated embedding methods to capture complex spatiotemporal dependencies can effectively address this problem. Motivated by these findings, we propose a novel embedding strategy that moderately

extracts features along the temporal dimension to balance prediction accuracy and computational complexity, while integrating multivariate features to enhance informative content, thereby enabling the efficient acquisition of high-quality embeddings. Combined with input inversion, this approach brings comprehensive improvements to Transformer-based forecasters.

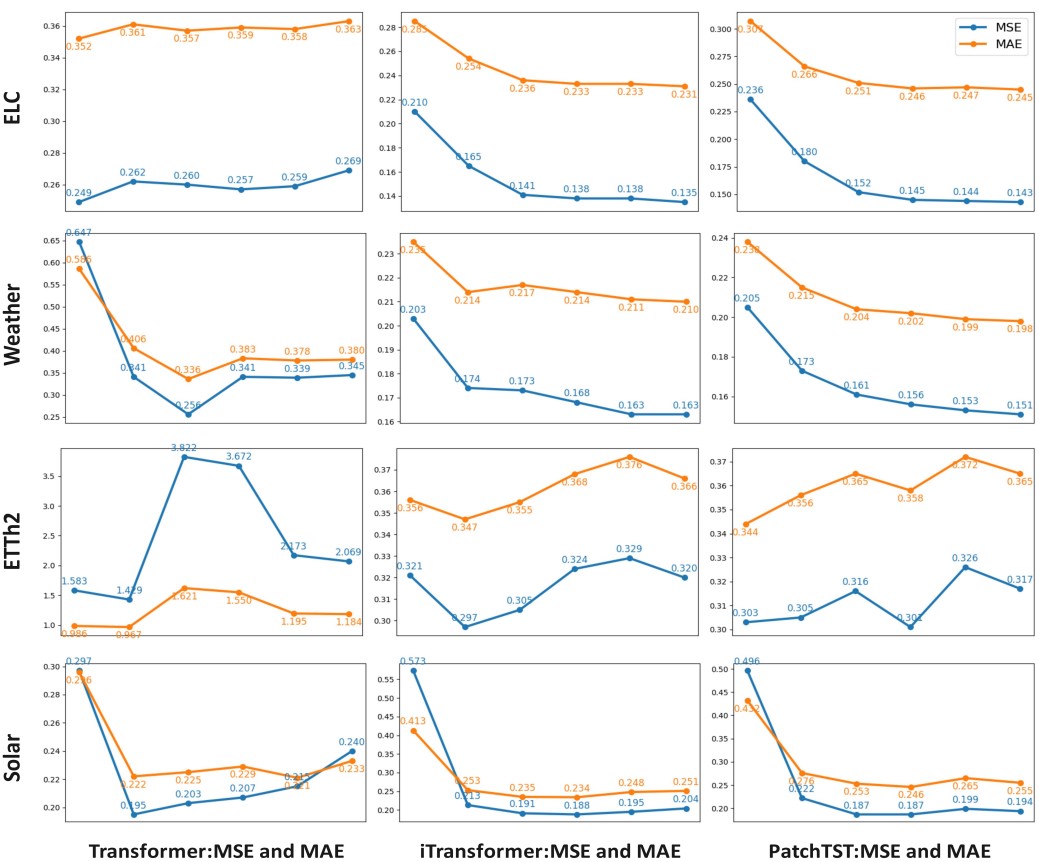

Figure 6: On the ECL, Weather, ETTh2, and Solar, we compared three models with input lengths of 48, 96, 144, 192, 240, 288 and an output length of 96. The evaluation metrics were MSE and MAE.

# B  LIMITATIONS OF CONVENTIONAL EMBEDDING

Time series data inherently exhibits both temporal and multivariate (spatial) characteristics. A growing body of research has demonstrated the effectiveness of explicitly capturing intra- and inter-channel dependencies (Zhang & Yan, 2023; Liu et al., 2024b;a). In prevailing Transformer-based forecasting frameworks, the embedding stage typically consists of three main components: timestamp embedding, positional embedding, and time series convolution. These components are primarily designed to capture temporal patterns while often overlooking the intrinsic correlations among different variables within the data. In fact, many types of time series data exhibit significant multivariate correlations, such as traffic flow and weather forecasting (Li et al., 2018; Wu et al., 2020b), where sensors are installed at fixed spatial locations. However, conventional embeddings capture only temporal patterns, while the subsequent attention layer also primarily models temporal dependencies, thereby overlooking multivariate dependencies and leading to information loss. Due to the overemphasis on temporal modeling and the loss of multivariate information under such designs, existing Transformer-based forecasters have encountered significant bottlenecks in capturing the contextual relationships essential for accurate multivariate time series forecasting.

Furthermore, the temporal feature extraction approach of the embedding layer, inspired by large language models (LLMs), is ill-suited for time series analysis. Unlike the Transformers used in natural language and vision tasks (Kirillov et al., 2023; OpenAI, 2023), which require learning dependencies among thousands to millions of tokens, Transformers in time series forecasting generally operate within relatively limited contexts, involving no more than hundreds of time series tokens. A smaller number of tokens makes it difficult for the attention mechanism to learn clear temporal correlations. However, time series convolution is limited to extracting local patterns from the original sequence and lacks the capability for fine-grained feature extraction. Additionally, PRformer (Yu et al., 2025) highlights that traditional Transformer algorithms attempt to learn sequence variation patterns by combining positional encoding with multivariate data at different time instants. However, real-world time series are frequently non-stationary, and the assumption of translation invariance makes it difficult for the Transformer's self-attention mechanism to learn correct temporal dependencies. Our experimental results (see Appendix A) indicate that the quality of the embeddings has a substantial impact on the performance of Transformer-based forecasters. Therefore, it is imperative to redesign the embedding model to capture more effective representations.

The HTME extractor discards positional encoding and treats timestamp information as a separate variable, concatenated to the time series inputs. This method has enhanced the model's ability to learn the relationship between the sequence and time. It employs patch-based operations and convolution to more effectively capture short-term dependencies, and a linear layer to capture long-term correlations. This method can extract temporal features more effectively and is better suited for Transformer-based architectures in time series forecasting tasks. We further employ a well-designed multivariate feature extractor to enrich the embedding representations with multivariate features. Notably, we do not explicitly model multivariate correlations, as doing so would incur substantial computational overhead. Finally, it incorporates multivariate features into temporal features to complement and adaptively balances the contributions of the two feature components, thereby enhancing the scalability of the embeddings. The resulting HTME incorporates rich temporal information and thoroughly accounts for multivariate correlation effects. It is more effective than existing embeddings in extracting informative representations from raw time series. This approach has already achieved good results in temporal feature extraction, making it unnecessary to overemphasize temporal correlations through the attention mechanism. So HTMformer utilizes inverted input to enable the attention mechanism to capture correlations among multiple features. iTransformer (Liu et al., 2024b) demonstrates that this approach leads to improved performance.

## C  TIME AND MEMORY COMPLEXITY

Table 6 provides a detailed comparison of the time complexity and memory usage across various Transformer-based forecasters. $L$ denotes the input sequence length, $N$ denotes the number of variables, and $S$ denotes the stride length of PatchTST.

Table 6: Time and memory complexity of different Transformer-based forecasters.

| Methods | Time | Memory |
|---|---|---|
| Transformer | $O(L^2)$ | $O(L^2)$ |
| Informer | $O(L \log L)$ | $O(L \log L)$ |
| Reformer | $O(L \log L)$ | $O(L \log L)$ |
| PatchTST | $O(L^2/S^2)$ | $O(L^2/S^2)$ |
| iTransformer | $O(N^2)$ | $O(N^2)$ |
| HTMformer | $O(N^2)$ | $O(N^2)$ |

## D    DETAILED EXPERIMENT SETTINGS

### D.1    DATASET DESCRIPTIONS

We perform extensive experiments on eight real-world datasets to evaluate the effectiveness of the proposed HTME strategy. These datasets cover diverse domains and temporal resolutions, including:

- **ECL.** Contains hourly electricity consumption data of 321 clients from 2012 to 2014.
- **Weather.** Includes 21 meteorological variables collected every 10 minutes in 2020 from the meteorological station at the Max Planck Institute for Biogeochemistry.
- **Traffic.** Consists of hourly road occupancy rates collected from 862 sensors on highways in the San Francisco Bay Area between January 2015 and December 2016.
- **ETTh.** Comprises hourly electricity transformer temperature readings collected over two years from two counties in China, with seven recorded variables.
- **Solar-Energy.** Records 10-minute solar power production records of 137 photovoltaic (PV) plants in Alabama from the year 2006.
- **PEMS.** Includes PEMS03, PEMS04, and PEMS08 datasets, each containing public transportation network data collected at 5-minute intervals in California.

To ensure the consistency of our experimental approach and make sure there are no data leakage issues, we split the ETT and PEMS datasets into the training, validation, and test sets, with a 6:2:2 ratio and other datasets with a 7:1:2 ratio, following the protocols recommended in recent literature (Wu et al., 2021; Zeng et al., 2023). In all forecasting experiments, the lookback window was fixed at 96 time steps, with prediction horizons of 96, 192, 336, 720 for the long-term experiments and 12, 24, 48, 96 for the short-term experiments. The details of all datasets are provided in Table 7.

Table 7: Detailed dataset descriptions. Dim denotes the number of variables in each dataset. Dataset Size indicates the number of time points in splits respectively. Prediction Length is the number of future time points to forecast. Frequency denotes the sampling interval of the time points.

| Dataset | Dim | Prediction Length | Dataset Size | Frequency | Forecastability∗ | Information |
|---------|-----|-------------------|--------------|-----------|------------------|-------------|
| ETTh2 | 7 | {96, 192, 336, 720} | (8545, 2881, 2881) | 15 min | 0.45 | Temperature |
| Electricity | 321 | {96, 192, 336, 720} | (18317, 2633, 5261) | Hourly | 0.77 | Electricity |
| Traffic | 862 | {96, 192, 336, 720} | (12185, 1757, 3509) | Hourly | 0.68 | Transportation |
| Weather | 21 | {96, 192, 336, 720} | (36792, 5271, 10540) | 10 min | 0.75 | Weather |
| Solar-Energy | 137 | {96, 192, 336, 720} | (36601, 5161, 10417) | 10 min | 0.33 | Solar Energy |
| PEMS03 | 358 | {12, 24, 48, 96, 192, 336, 720} | (15617, 5135, 5135) | 5 min | 0.65 | Transportation |
| PEMS04 | 307 | {12, 24, 48, 96, 192, 336, 720} | (10172, 3375, 3375) | 5 min | 0.45 | Transportation |
| PEMS08 | 170 | {12, 24, 48, 96, 192, 336, 720} | (10690, 3548, 265) | 5 min | 0.52 | Transportation |

∗ The forecastability is calculated by one minus the entropy of Fourier decomposition of time series (Goerg, 2013). A larger value indicates better predictability.

### D.2    BASELINE DETAILS

We employ five state-of-the-art Transformer-based models, such as iTransformer (Liu et al., 2024b), PatchTST (Nie et al., 2023), FEDformer (Zhou et al., 2022), WPMixer (Murad et al., 2025), and MultiPatchFormer (Naghashi et al., 2025), as well as three well-acknowledged non-Transformer-based models, Dlinear (Zeng et al., 2023), TimeMixer (Wang et al., 2024a), and SegRNN (Lin et al., 2023). We utilized the official repositories of the Time-Series-Library (Wang et al., 2024b) directly for reproduction, and adopted the same comparison settings. The Time-Series-Library consolidates and integrates implementations of prior time-series forecasting models, making it convenient for us to conduct experiments. Among these models, MultiPatchFormer is a highly advanced architecture and yields the best overall forecasting accuracy. Both iTransformer and our model utilize input inversion to enable the attention mechanism to effectively capture multivariate correlations. WPMixer efficiently

captures both the frequency-domain and time-domain information in time series data, enabling deep temporal feature extraction. FEDformer achieves high forecasting accuracy and computational efficiency through an innovative architecture that integrates seasonal-trend decomposition with frequency-domain enhancement. Meanwhile, DLinear is a state-of-the-art linear model with a non-Transformer architecture. SegRNN has demonstrated outstanding performance on multiple benchmark time series forecasting datasets by employing segment-wise iterative processing and parallelized multi-step prediction. TimeMixer is a linear time series forecasting model that leverages a multi-scale decomposable mixing mechanism, demonstrating robust performance on high-dimensional datasets. Note that SegRNN is incompatible with short-term forecasting, so no comparison is conducted.

### D.3 EXPERIMENT DETAILS

All experiments were implemented in PyTorch (Paszke et al., 2019) and conducted on a dedicated NVIDIA Quadro RTX 6000 GPU. We trained the model for 10 epochs using the Adam optimizer (Kingma & Ba, 2015), with an early stopping mechanism configured at a patience value of 3. The key parameter configurations are listed in Table 8 (WPMixer is configured with d_model = 16 and d_ff = 32). We verified that, for these baselines, all other hyperparameters were taken from their official repositories within the Time-Series-Library (Wang et al., 2024b), which consolidates implementations of prior time-series forecasting models. This ensures consistency with the established fair comparison protocol. Unless otherwise stated, all models adopt the aforementioned configurations across all experiments. The only modification was to the output sequence lengths.

- **dropout**: Dropout rate.
- **batch size**: Batch size used for training.
- **l_rate**: The learning rate used in the optimization process.
- **d_ff**: Dimension of Fully FFN
- **d_model**: Dimension of model
- **e_layers**: Number of Transformer encoder layers.
- **n_heads**: Num of heads
- **seq_len**: Inputs sequence length
- **label_len**: Start token length
- **loss**: Loss function used for training

Table 8: Experiments of all models on all datasets follow the following experimental settings to ensure the accuracy of the experiments.

| dropout | batch size | l_rate | d_ff | d_model | e_layers | n_heads | seq_lenl | label_len | loss |
|---------|-----------|--------|------|---------|----------|---------|----------|-----------|------|
| 0.1 | 32 | 0.0001 | 1024 | 512 | 2 | 8 | 96 | 48 | L2 |

## E  FULL RESULTS

### E.1 PERFORMANCE PROMOTION WITH HTME

To assess the effectiveness and scalability of the HTME for enhancing the Transformer's capacity to learn temporal sequence representations, we integrated HTME into the Transformer and its variants: Transformer (Vaswani et al., 2017), Reformer (Kitaev et al., 2020), Informer (Zhou et al., 2021), Flowformer (Ma et al., 2023), and Flashformer (Dao et al., 2022). The length of historical data sequences was set to 96 time steps, with the forecast horizon varying among 96, 192, 336, and 720 time steps. The full predictive outcomes are detailed in Table 9. On three benchmark datasets (ECL, Traffic, and Weather), the HTME versions yield significant improvements over all four models across all proposed prediction horizons, thereby confirming their effectiveness and scalability. We further integrated the multivariate feature extraction module from the HTME extractor into iTransformer to assess the scalability capability of this module, with the detailed results presented in the ablation study. This component was seamlessly incorporated into the iTransformer's embedding layer, leading to a substantial improvement in its overall performance.

Table 9: Full results of original Transformers and Transformers with HTME.

| Models | | | Transformer (2017) | | Reformer 2020 | | Informer 2021 | | Flowformer 2023 | | Flashformer 2022 | |
|---|---|---|---|---|---|---|---|---|---|---|---|---|
| Dataset | Type | Step | MSE | MAE | MSE | MAE | MSE | MAE | MSE | MAE | MSE | MAE |
| Electricity | Original | 96 | 0.264 | 0.363 | 0.304 | 0.391 | 0.330 | 0.415 | 0.272 | 0.371 | 0.263 | 0.361 |
| | | 192 | 0.275 | 0.375 | 0.350 | 0.426 | 0.356 | 0.438 | 0.269 | 0.371 | 0.270 | 0.369 |
| | | 336 | 0.262 | 0.362 | 0.354 | 0.429 | 0.376 | 0.454 | 0.264 | 0.364 | 0.263 | 0.360 |
| | | 720 | 0.286 | 0.373 | 0.361 | 0.422 | 0.454 | 0.464 | 0.284 | 0.374 | 0.279 | 0.368 |
| | | Avg | 0.271 | 0.363 | 0.342 | 0.417 | 0.379 | 0.442 | 0.272 | 0.370 | 0.268 | 0.364 |
| | +HTME | 96 | 0.157 | 0.247 | 0.169 | 0.266 | 0.170 | 0.266 | 0.170 | 0.266 | 0.163 | 0.262 |
| | | 192 | 0.171 | 0.260 | 0.190 | 0.283 | 0.187 | 0.279 | 0.188 | 0.280 | 0.183 | 0.277 |
| | | 336 | 0.192 | 0.277 | 0.200 | 0.293 | 0.207 | 0.296 | 0.202 | 0.294 | 0.199 | 0.292 |
| | | 720 | 0.220 | 0.304 | 0.248 | 0.329 | 0.247 | 0.329 | 0.235 | 0.319 | 0.229 | 0.316 |
| | | Avg | 0.185 | 0.272 | 0.201 | 0.292 | 0.202 | 0.292 | 0.198 | 0.289 | 0.194 | 0.287 |
| Weather | Original | 96 | 0.372 | 0.417 | 0.311 | 0.355 | 0.420 | 0.452 | 0.283 | 0.364 | 0.326 | 0.395 |
| | | 192 | 0.632 | 0.580 | 0.453 | 0.464 | 0.423 | 0.455 | 0.548 | 0.541 | 0.587 | 0.544 |
| | | 336 | 0.699 | 0.606 | 0.476 | 0.474 | 0.588 | 0.539 | 0.614 | 0.548 | 0.714 | 0.614 |
| | | 720 | 0.903 | 0.688 | 0.579 | 0.541 | 1.146 | 0.784 | 1.079 | 0.783 | 0.906 | 0.684 |
| | | Avg | 0.651 | 0.572 | 0.454 | 0.458 | 0.644 | 0.557 | 0.631 | 0.559 | 0.633 | 0.559 |
| | +HTME | 96 | 0.163 | 0.209 | 0.164 | 0.209 | 0.170 | 0.213 | 0.167 | 0.212 | 0.168 | 0.215 |
| | | 192 | 0.222 | 0.261 | 0.214 | 0.255 | 0.220 | 0.260 | 0.216 | 0.255 | 0.218 | 0.258 |
| | | 336 | 0.272 | 0.290 | 0.270 | 0.288 | 0.274 | 0.298 | 0.279 | 0.302 | 0.275 | 0.298 |
| | | 720 | 0.359 | 0.352 | 0.351 | 0.348 | 0.354 | 0.351 | 0.354 | 0.350 | 0.352 | 0.349 |
| | | Avg | 0.254 | 0.277 | 0.249 | 0.275 | 0.254 | 0.280 | 0.254 | 0.279 | 0.253 | 0.380 |
| Traffic | Original | 96 | 0.658 | 0.364 | 0.715 | 0.399 | 0.731 | 0.413 | 0.646 | 0.359 | 0.637 | 0.353 |
| | | 192 | 0.655 | 0.359 | 0.705 | 0.390 | 0.761 | 0.431 | 0.646 | 0.355 | 0.659 | 0.364 |
| | | 336 | 0.657 | 0.361 | 0.703 | 0.386 | 0.840 | 0.474 | 0.657 | 0.358 | 0.653 | 0.357 |
| | | 720 | 0.697 | 0.380 | 0.703 | 0.383 | 0.969 | 0.545 | 0.671 | 0.364 | 0.697 | 0.375 |
| | | Avg | 0.666 | 0.366 | 0.706 | 0.389 | 0.825 | 0.465 | 0.655 | 0.359 | 0.661 | 0.362 |
| | +HTME | 96 | 0.439 | 0.300 | 0.445 | 0.302 | 0.466 | 0.314 | 0.451 | 0.309 | 0.439 | 0.300 |
| | | 192 | 0.450 | 0.302 | 0.463 | 0.308 | 0.468 | 0.312 | 0.466 | 0.310 | 0.460 | 0.310 |
| | | 336 | 0.462 | 0.310 | 0.478 | 0.313 | 0.488 | 0.322 | 0.484 | 0.320 | 0.480 | 0.321 |
| | | 720 | 0.517 | 0.339 | 0.514 | 0.334 | 0.530 | 0.347 | 0.524 | 0.345 | 0.516 | 0.342 |
| | | Avg | 0.467 | 0.312 | 0.475 | 0.314 | 0.488 | 0.323 | 0.481 | 0.321 | 0.474 | 0.319 |

### E.2 LONG-TERM FORECASTING RESULTS

The results of the multivariate long-term forecasting tasks are summarized in Table 10. Lower Mean Squared Error (MSE) and Mean Absolute Error (MAE) values indicate superior predictive accuracy. The best results are highlighted in **red**, and the second-best results in blue. Notably, HTMformer achieves state-of-the-art performance in most evaluation scenarios, ranking first in 26 out of 40 MSE metrics and 29 out of 40 MAE metrics. Moreover, our model consistently ranks within the top two across the majority of evaluation scenarios. Table 11 also reports the standard deviations of HTMformer over five runs with different random seeds, demonstrating its performance stability.

### E.3 SHORT-TERM FORECASTING RESULTS

Table 12 presents the short-term forecasting results. The evaluation metrics comprise the Mean Squared Error (MSE) and Mean Absolute Error (MAE) computed over four prediction horizons. Our approach consistently achieves substantial improvements across all horizons on the selected datasets. In particular, on the PEMS04 dataset, our proposed model achieved a *33.3%* reduction in average MSE and an *18.3%* reduction in average MAE compared to the second-best results. The standard deviations, computed over five independent runs with different random seeds, are reported in Table 13. For short-term forecasting, the limited length of the input sequence constrains the model's capacity to capture sufficient temporal dependencies. Consequently, modeling multivariate correlations becomes crucial, in which HTMformer demonstrates superior performance.

Table 10: Full results for the long-term forecasting task. We compare extensive competitive models under different prediction lengths. *Avg* is averaged from all four prediction lengths.

| | Models | HTMformer (Ours) | | iTransformer (2024b) | | PatchTST 2023 | | DLinear 2023 | | FEDformer 2022 | | MPFormer 2025 | | WPMixer 2025 | | TimeMixer 2024a | | SegRNN 2023 | |
|---|---|---|---|---|---|---|---|---|---|---|---|---|---|---|---|---|---|---|---|
| | Metric | MSE | MAE | MSE | MAE | MSE | MAE | MSE | MAE | MSE | MAE | MSE | MAE | MSE | MAE | MSE | MAE | MSE | MAE |
| Electricity | 96 | **0.157** | **0.247** | 0.163 | 0.252 | 0.180 | 0.266 | 0.210 | 0.301 | 0.195 | 0.309 | 0.159 | 0.249 | 0.170 | 0.259 | 0.213 | 0.301 | 0.191 | 0.278 |
| | 192 | **0.171** | **0.260** | 0.175 | 0.263 | 0.185 | 0.271 | 0.210 | 0.304 | 0.202 | 0.315 | 0.171 | 0.260 | 0.180 | 0.267 | 0.225 | 0.316 | 0.202 | 0.287 |
| | 336 | 0.192 | 0.277 | 0.282 | 0.299 | 0.202 | 0.288 | 0.223 | 0.319 | 0.229 | 0.342 | 0.188 | 0.277 | 0.196 | 0.284 | 0.242 | 0.334 | 0.221 | 0.305 |
| | 720 | **0.220** | **0.304** | 0.357 | 0.348 | 0.241 | 0.319 | 0.257 | 0.349 | 0.264 | 0.367 | 0.228 | 0.309 | 0.238 | 0.317 | 0.285 | 0.363 | 0.261 | 0.339 |
| | Avg | **0.185** | **0.272** | 0.244 | 0.290 | 0.202 | 0.286 | 0.225 | 0.318 | 0.222 | 0.333 | 0.186 | 0.273 | 0.196 | 0.281 | 0.241 | 0.328 | 0.218 | 0.302 |
| Weather | 96 | **0.164** | **0.209** | 0.176 | 0.216 | 0.174 | 0.217 | 0.196 | 0.256 | 0.224 | 0.304 | 0.168 | 0.209 | 0.165 | 0.209 | 0.179 | 0.225 | 0.167 | 0.230 |
| | 192 | 0.222 | 0.261 | 0.223 | 0.255 | 0.220 | 0.256 | 0.238 | 0.299 | 0.281 | 0.348 | 0.213 | 0.250 | 0.210 | 0.251 | 0.230 | 0.268 | 0.214 | 0.275 |
| | 336 | 0.272 | **0.290** | 0.280 | 0.298 | 0.276 | 0.296 | 0.281 | 0.330 | 0.339 | 0.381 | 0.273 | 0.293 | 0.266 | 0.291 | 0.284 | 0.304 | 0.271 | 0.317 |
| | 720 | 0.359 | 0.352 | 0.353 | 0.346 | 0.353 | 0.346 | **0.345** | 0.381 | 0.408 | 0.417 | 0.354 | 0.347 | 0.345 | 0.344 | 0.356 | 0.350 | 0.360 | 0.378 |
| | Avg | 0.254 | 0.277 | 0.259 | 0.279 | 0.255 | 0.278 | 0.265 | 0.316 | 0.313 | 0.362 | 0.252 | 0.274 | 0.246 | 0.273 | 0.262 | 0.286 | 0.253 | 0.300 |
| Traffic | 96 | 0.439 | 0.300 | 0.442 | 0.302 | 0.494 | 0.313 | 0.696 | 0.428 | 0.580 | 0.362 | 0.433 | 0.290 | 0.516 | 0.336 | 0.688 | 0.429 | 0.781 | 0.401 |
| | 192 | 0.450 | 0.302 | 0.459 | 0.308 | 0.490 | 0.330 | 0.646 | 0.407 | 0.606 | 0.379 | 0.450 | 0.296 | 0.512 | 0.331 | 0.687 | 0.434 | 0.789 | 0.405 |
| | 336 | 0.462 | 0.310 | 0.479 | 0.319 | 0.502 | 0.317 | 0.653 | 0.409 | 0.613 | 0.380 | 0.467 | 0.306 | 0.353 | 0.334 | 0.719 | 0.451 | 0.797 | 0.405 |
| | 720 | 0.517 | 0.339 | 0.516 | 0.342 | 0.545 | 0.337 | 0.694 | 0.428 | 0.641 | 0.393 | 0.500 | 0.326 | 0.558 | 0.353 | 0.761 | 0.466 | 0.813 | 0.421 |
| | Avg | 0.467 | 0.312 | 0.474 | 0.317 | 0.507 | 0.324 | 0.672 | 0.418 | 0.610 | 0.378 | 0.462 | 0.304 | 0.484 | 0.338 | 0.713 | 0.445 | 0.795 | 0.408 |
| ETTh2 | 96 | 0.300 | 0.348 | 0.295 | 0.344 | 0.293 | 0.343 | 0.341 | 0.395 | 0.350 | 0.391 | 0.299 | 0.350 | 0.293 | 0.345 | 0.296 | 0.345 | 0.295 | 0.355 |
| | 192 | 0.389 | 0.402 | 0.375 | 0.398 | 0.373 | 0.393 | 0.481 | 0.479 | 0.441 | 0.449 | 0.384 | 0.401 | 0.382 | 0.401 | 0.384 | 0.401 | 0.375 | 0.397 |
| | 336 | 0.409 | 0.408 | 0.436 | 0.435 | 0.422 | 0.430 | 0.592 | 0.542 | 0.498 | 0.490 | 0.411 | 0.428 | 0.424 | 0.437 | 0.429 | 0.439 | 0.414 | 0.436 |
| | 720 | 0.421 | 0.439 | 0.429 | 0.447 | 0.440 | 0.452 | 0.840 | 0.661 | 0.480 | 0.487 | 0.430 | 0.446 | 0.449 | 0.458 | 0.433 | 0.451 | 0.443 | 0.469 |
| | Avg | 0.379 | 0.399 | 0.383 | 0.406 | 0.382 | 0.404 | 0.563 | 0.519 | 0.442 | 0.454 | 0.381 | 0.406 | 0.387 | 0.410 | 0.385 | 0.409 | 0.381 | 0.414 |
| Solar-Energy | 96 | **0.199** | **0.239** | 0.213 | 0.253 | 0.214 | 0.257 | 0.289 | 0.377 | 0.279 | 0.363 | 0.203 | 0.240 | 0.232 | 0.278 | 0.256 | 0.292 | 0.232 | 0.294 |
| | 192 | **0.233** | **0.263** | 0.242 | 0.274 | 0.254 | 0.296 | 0.319 | 0.397 | 0.288 | 0.378 | 0.237 | 0.268 | 0.268 | 0.303 | 0.286 | 0.315 | 0.253 | 0.306 |
| | 336 | 0.252 | **0.279** | 0.262 | 0.290 | 0.284 | 0.314 | 0.352 | 0.415 | 0.316 | 0.399 | 0.249 | 0.280 | 0.291 | 0.316 | 0.323 | 0.337 | 0.264 | 0.310 |
| | 720 | 0.259 | **0.280** | 0.270 | 0.296 | 0.263 | 0.291 | 0.356 | 0.412 | 0.363 | 0.430 | 0.254 | 0.283 | 0.290 | 0.315 | 0.346 | 0.349 | 0.260 | 0.306 |
| | Avg | **0.235** | **0.265** | 0.246 | 0.278 | 0.253 | 0.289 | 0.329 | 0.400 | 0.311 | 0.392 | 0.235 | 0.267 | 0.270 | 0.303 | 0.302 | 0.323 | 0.252 | 0.304 |
| PEMS03 | 96 | **0.25** | **0.343** | 0.337 | 0.413 | 0.504 | 0.516 | 0.458 | 0.517 | 0.325 | 0.423 | 0.302 | 0.395 | 0.514 | 0.505 | 0.731 | 0.625 | 0.381 | 0.438 |
| | 192 | **0.297** | **0.378** | 0.374 | 0.436 | 0.547 | 0.544 | 0.477 | 0.527 | 0.475 | 0.515 | 0.349 | 0.428 | 0.557 | 0.526 | 0.797 | 0.648 | 0.447 | 0.478 |
| | 336 | **0.274** | **0.356** | 0.332 | 0.395 | 0.432 | 0.460 | 0.396 | 0.456 | 0.425 | 0.480 | 0.313 | 0.388 | 0.440 | 0.450 | 0.614 | 0.545 | 0.380 | 0.432 |
| | 720 | **0.336** | **0.401** | 0.397 | 0.440 | 0.527 | 0.521 | 0.439 | 0.492 | 0.644 | 0.596 | 0.383 | 0.441 | 0.533 | 0.509 | 0.762 | 0.638 | 0.447 | 0.478 |
| | Avg | **0.289** | **0.369** | 0.360 | 0.421 | 0.502 | 0.510 | 0.442 | 0.498 | 0.467 | 0.503 | 0.336 | 0.413 | 0.511 | 0.497 | 0.726 | 0.614 | 0.413 | 0.456 |
| PEMS04 | 96 | **0.226** | **0.335** | 0.387 | 0.446 | 0.645 | 0.594 | 0.452 | 0.504 | 0.300 | 0.411 | 0.356 | 0.428 | 0.584 | 0.554 | 0.794 | 0.658 | 0.415 | 0.468 |
| | 192 | **0.286** | **0.383** | 0.429 | 0.473 | 0.688 | 0.615 | 0.477 | 0.527 | 0.458 | 0.513 | 0.398 | 0.459 | 0.650 | 0.586 | 0.891 | 0.697 | 0.496 | 0.510 |
| | 336 | **0.286** | **0.372** | 0.369 | 0.426 | 0.536 | 0.516 | 0.396 | 0.456 | 0.411 | 0.474 | 0.348 | 0.413 | 0.511 | 0.499 | 0.703 | 0.596 | 0.420 | 0.461 |
| | 720 | **0.338** | **0.414** | 0.440 | 0.476 | 0.629 | 0.578 | 0.439 | 0.492 | 0.718 | 0.630 | 0.423 | 0.467 | 0.601 | 0.556 | 0.812 | 0.660 | 0.479 | 0.502 |
| | Avg | **0.284** | **0.376** | 0.406 | 0.455 | 0.624 | 0.575 | 0.441 | 0.494 | 0.471 | 0.507 | 0.381 | 0.441 | 0.586 | 0.548 | 0.800 | 0.652 | 0.452 | 0.485 |
| PEMS08 | 96 | **0.365** | **0.409** | 0.477 | 0.470 | 0.567 | 0.534 | 0.672 | 0.564 | 0.504 | 0.506 | 0.406 | 0.437 | 0.648 | 0.556 | 0.854 | 0.657 | 0.516 | 0.487 |
| | 192 | **0.513** | **0.462** | 0.627 | 0.518 | 0.794 | 0.620 | 0.726 | 0.580 | 0.726 | 0.608 | 0.539 | 0.479 | 0.766 | 0.581 | 1.048 | 0.702 | 0.646 | 0.528 |
| | 336 | **0.530** | **0.435** | 0.599 | 0.472 | 0.709 | 0.539 | 0.650 | 0.518 | 0.803 | 0.615 | 0.538 | 0.444 | 0.696 | 0.518 | 0.906 | 0.615 | 0.617 | 0.490 |
| | 720 | 0.624 | **0.487** | 0.691 | 0.532 | 0.807 | 0.599 | 0.713 | 0.562 | 1.000 | 0.711 | 0.603 | 0.489 | 0.789 | 0.575 | 1.019 | 0.682 | 0.707 | 0.541 |
| | Avg | **0.508** | **0.448** | 0.598 | 0.498 | 0.719 | 0.573 | 0.690 | 0.556 | 0.758 | 0.610 | 0.521 | 0.462 | 0.724 | 0.557 | 0.956 | 0.664 | 0.621 | 0.511 |
| | Count | 26 | 29 | 0 | 0 | 2 | 2 | 0 | 0 | 0 | 0 | 10 | 9 | 6 | 3 | 0 | 0 | 0 | 0 |

Table 11: Robustness of HTMformer performance is evaluated over five random seeds.

| Dataset | ECL | | Weather | | Traffic | | ETTh2 | |
|---|---|---|---|---|---|---|---|---|
| Horizon | MSE | MAE | MSE | MAE | MSE | MAE | MSE | MAE |
| 96 | $0.157 \pm 0.003$ | $0.247 \pm 0.002$ | $0.164 \pm 0.004$ | $0.209 \pm 0.002$ | $0.439 \pm 0.002$ | $0.300 \pm 0.002$ | $0.300 \pm 0.003$ | $0.348 \pm 0.004$ |
| 192 | $0.171 \pm 0.003$ | $0.260 \pm 0.002$ | $0.222 \pm 0.005$ | $0.261 \pm 0.004$ | $0.450 \pm 0.002$ | $0.302 \pm 0.003$ | $0.389 \pm 0.004$ | $0.402 \pm 0.003$ |
| 336 | $0.192 \pm 0.005$ | $0.277 \pm 0.005$ | $0.272 \pm 0.005$ | $0.290 \pm 0.010$ | $0.462 \pm 0.004$ | $0.310 \pm 0.005$ | $0.409 \pm 0.004$ | $0.408 \pm 0.004$ |
| 720 | $0.220 \pm 0.006$ | $0.304 \pm 0.007$ | $0.359 \pm 0.004$ | $0.352 \pm 0.002$ | $0.517 \pm 0.009$ | $0.339 \pm 0.007$ | $0.421 \pm 0.007$ | $0.439 \pm 0.007$ |

| Dataset | Solar-Energy | | PEMS03 | | PEMS04 | | PEMS08 | |
|---|---|---|---|---|---|---|---|---|
| Horizon | MSE | MAE | MSE | MAE | MSE | MAE | MSE | MAE |
| 96 | $0.199 \pm 0.004$ | $0.239 \pm 0.003$ | $0.250 \pm 0.003$ | $0.343 \pm 0.002$ | $0.226 \pm 0.008$ | $0.335 \pm 0.004$ | $0.365 \pm 0.003$ | $0.409 \pm 0.005$ |
| 192 | $0.233 \pm 0.004$ | $0.263 \pm 0.006$ | $0.297 \pm 0.005$ | $0.378 \pm 0.010$ | $0.286 \pm 0.010$ | $0.383 \pm 0.005$ | $0.513 \pm 0.005$ | $0.462 \pm 0.006$ |
| 336 | $0.252 \pm 0.006$ | $0.279 \pm 0.007$ | $0.274 \pm 0.010$ | $0.356 \pm 0.009$ | $0.286 \pm 0.010$ | $0.372 \pm 0.005$ | $0.530 \pm 0.006$ | $0.435 \pm 0.010$ |
| 720 | $0.259 \pm 0.006$ | $0.280 \pm 0.009$ | $0.336 \pm 0.007$ | $0.401 \pm 0.008$ | $0.338 \pm 0.009$ | $0.414 \pm 0.007$ | $0.624 \pm 0.003$ | $0.487 \pm 0.006$ |

Table 12: Full results for the short-term forecasting task. We compare extensive competitive models on PEMS datasets. *Avg* means the average results from all four prediction lengths.

| Models | HTMformer (Ours) | | iTransformer (2024b) | | PatchTST 2023 | | DLinear 2023 | | FEDformer 2022 | | MPFormer 2025 | | WPMixer 2025 | | TimeMixer 2024a | |
|---|---|---|---|---|---|---|---|---|---|---|---|---|---|---|---|---|
| Metric | MSE | MAE | MSE | MAE | MSE | MAE | MSE | MAE | MSE | MAE | MSE | MAE | MSE | MAE | MSE | MAE |
| PEMS03 12 | **0.067** | **0.174** | 0.075 | 0.184 | 0.102 | 0.217 | 0.122 | 0.245 | 0.125 | 0.251 | 0.089 | 0.199 | 0.093 | 0.204 | 0.107 | 0.220 |
| PEMS03 24 | **0.093** | **0.204** | 0.115 | 0.229 | 0.187 | 0.298 | 0.201 | 0.320 | 0.152 | 0.279 | 0.114 | 0.225 | 0.151 | 0.263 | 0.185 | 0.292 |
| PEMS03 48 | **0.151** | **0.261** | 0.195 | 0.305 | 0.278 | 0.369 | 0.334 | 0.428 | 0.222 | 0.347 | 0.186 | 0.296 | 0.292 | 0.373 | 0.375 | 0.428 |
| PEMS03 96 | **0.241** | **0.341** | 0.340 | 0.417 | 0.504 | 0.516 | 0.458 | 0.517 | 0.325 | 0.423 | 0.300 | 0.393 | 0.514 | 0.505 | 0.731 | 0.625 |
| PEMS03 Avg | **0.138** | **0.245** | 0.181 | 0.283 | 0.267 | 0.350 | 0.278 | 0.377 | 0.206 | 0.325 | 0.172 | 0.278 | 0.262 | 0.336 | 0.348 | 0.391 |
| PEMS04 12 | **0.076** | **0.183** | 0.095 | 0.202 | 0.112 | 0.231 | 0.147 | 0.272 | 0.136 | 0.263 | 0.109 | 0.218 | 0.111 | 0.223 | 0.126 | 0.239 |
| PEMS04 24 | **0.097** | **0.209** | 0.140 | 0.249 | 0.187 | 0.301 | 0.224 | 0.340 | 0.156 | 0.284 | 0.138 | 0.248 | 0.181 | 0.289 | 0.208 | 0.312 |
| PEMS04 48 | **0.146** | **0.263** | 0.238 | 0.333 | 0.355 | 0.422 | 0.356 | 0.437 | 0.226 | 0.351 | 0.226 | 0.326 | 0.343 | 0.411 | 0.422 | 0.456 |
| PEMS04 96 | **0.233** | **0.342** | 0.394 | 0.449 | 0.638 | 0.587 | 0.453 | 0.504 | 0.308 | 0.417 | 0.358 | 0.430 | 0.584 | 0.554 | 0.794 | 0.658 |
| PEMS04 Avg | **0.138** | **0.249** | 0.216 | 0.308 | 0.323 | 0.385 | 0.295 | 0.388 | 0.206 | 0.328 | 0.207 | 0.305 | 0.304 | 0.369 | 0.387 | 0.426 |
| PEMS07 12 | **0.081** | **0.187** | 0.086 | 0.190 | 0.108 | 0.225 | 0.152 | 0.274 | 0.175 | 0.273 | 0.099 | 0.205 | 0.103 | 0.212 | 0.117 | 0.230 |
| PEMS07 24 | **0.120** | **0.227** | 0.135 | 0.240 | 0.173 | 0.284 | 0.246 | 0.351 | 0.212 | 0.307 | 0.127 | 0.234 | 0.174 | 0.279 | 0.196 | 0.302 |
| PEMS07 48 | **0.198** | **0.297** | 0.247 | 0.333 | 0.341 | 0.404 | 0.438 | 0.469 | 0.296 | 0.375 | 0.219 | 0.312 | 0.332 | 0.396 | 0.402 | 0.443 |
| PEMS07 96 | **0.352** | **0.401** | 0.486 | 0.478 | 0.567 | 0.534 | 0.672 | 0.564 | 0.463 | 0.481 | 0.410 | 0.434 | 0.648 | 0.556 | 0.854 | 0.657 |
| PEMS07 Avg | **0.187** | **0.278** | 0.238 | 0.310 | 0.297 | 0.361 | 0.377 | 0.414 | 0.286 | 0.359 | 0.213 | 0.296 | 0.314 | 0.360 | 0.392 | 0.408 |
| Count | **15** | **15** | 0 | 0 | 0 | 0 | 0 | 0 | 0 | 0 | 0 | 0 | 0 | 0 | 0 | 0 |

Table 13: Results on short-term time series forecasting are obtained from five random seeds.

| Dataset | PEMS03 | | PEMS04 | | PEMS08 | |
|---|---|---|---|---|---|---|
| Horizon | MSE | MAE | MSE | MAE | MSE | MAE |
| 12 | $0.067 \pm 0.000$ | $0.174 \pm 0.000$ | $0.076 \pm 0.000$ | $0.183 \pm 0.000$ | $0.081 \pm 0.002$ | $0.187 \pm 0.002$ |
| 24 | $0.093 \pm 0.001$ | $0.204 \pm 0.000$ | $0.097 \pm 0.001$ | $0.209 \pm 0.002$ | $0.120 \pm 0.001$ | $0.227 \pm 0.001$ |
| 48 | $0.151 \pm 0.003$ | $0.261 \pm 0.001$ | $0.146 \pm 0.004$ | $0.263 \pm 0.004$ | $0.198 \pm 0.002$ | $0.297 \pm 0.002$ |
| 96 | $0.241 \pm 0.000$ | $0.341 \pm 0.003$ | $0.233 \pm 0.005$ | $0.342 \pm 0.005$ | $0.352 \pm 0.006$ | $0.401 \pm 0.002$ |

# F  ADDITIONAL ABLATION STUDIES

To investigate the functional rationale of HTMformer components, we conduct detailed ablation experiments involving both component replacement and component removal. The detailed results are reported in Table 14 of the main text. As all experiments were conducted under identical configurations, the HTMformer variants are directly comparable to other models.

**Effect of the temporal feature extraction module.**  By comparing iTransformer and HTMformerV1, blue numbers are used to denote cases where HTMformerV1 outperforms iTransformer. Observations indicate that HTMformerV1 demonstrates improvements in 63/80 cases. By comparing HTMformerV1 and HTMformerV2, numbers prefixed with an underline indicate that HTMformerV1 outperforms HTMformerV2. Notably, HTMformerV1 performs better in 57/80 cases. Significantly, on Solar and PEMS datasets lacking timestamps, HTMformerV2 often exhibits superior performance, verifying our first hypothesis: 1) The majority of time series features are stored in the temporal dimension, and the depth of temporal feature mining substantially influences prediction accuracy.

**Effect of the multivariate features.**  Using iTransformer and HTMformerV1 as baselines, we compare them against iTransformerV3 and HTMformer versions. Red highlighted numbers indicate cases where the multivariate-integrated versions outperform their original counterparts. Experimental results show that iTransformerV3 achieves superior performance in 61/80 cases compared to iTransformer, while HTMformer outperforms its baseline in 68/80 cases. The gains are particularly evident on the PEMS datasets, which are derived from public transportation network data characterized by strong correlations among sensor nodes. This finding corroborates our second hypothesis: 2) The spatial dimension (multivariate dimension) also contains abundant features, and comprehensive multivariate feature mining can further reduce errors.

**Effect of the hybrid strategy.**  Among all models and their variants, HTMformer excels in 68/80 cases. This demonstrates the effectiveness of the combination. In most cases, temporal features are the primary characteristics of time series data, but the correlations between multiple variables cannot be overlooked. It is of great significance for improving the accuracy of Transformer-based predictors. Different datasets vary in the depth of temporal and multivariate features. Therefore, we use a learnable parameter to control the proportion of the two modules, allowing the fusion of multivariate features to still reduce errors to an extent. In conclusion, the strategy of fusing temporal features and multivariate features is superior to using either of them alone. This strategy can enhance the accuracy and scalability of Transformer-based predictors.

# G  DETAILS OF MODEL EFFICIENCY

For a comprehensive efficiency comparison, we evaluate HTMformer against three highly competitive baselines, including MultiPatchFormer (Naghashi et al., 2025), WPMixer (Murad et al., 2025), and PatchTST (Nie et al., 2023), across eight datasets. The evaluation considers three key metrics: training time, GPU memory footprint, and total parameter count, with the input sequence length fixed at 96 and the prediction length at 192. Moreover, the experimental hardware conditions and parameter configurations follow those described in Appendix B, and the best results are marked in red. The full results are shown in Table 15 (**Left**) and Table 15 (**Right**). The training time and GPU usage serve as an intuitive indicator of total parameter count, so detailed reporting is omitted. Across all datasets, HTMformer achieves the best training time and GPU memory footprint. Notably, for high-dimensional datasets, particularly when the number of variate dimensions exceeds one hundred, HTMformer delivers nearly three times the training speed of the second-best model and requires only one-third of its memory footprint. Moreover, this advantage becomes increasingly pronounced as dimensionality grows. This superiority stems from our lightweight design philosophy, in which HTME avoids overemphasizing temporal feature extraction and HTMformer adopts only a vanilla Transformer encoder combined with the inverted input strategy.

Table 14: We compare different variants under different prediction lengths on multiple datasets (MSE and MAE). The input sequence length is set to 96.

| Models | | HTMformer | | HTMformerV1 | | HTMformerV2 | | iTransformer | | iTransV3 | | MPFormer | |
|---|---|---|---|---|---|---|---|---|---|---|---|---|---|
| Metric | | MSE | MAE | MSE | MAE | MSE | MAE | MSE | MAE | MSE | MAE | MSE | MAE |
| ECL | 96 | 0.157 | 0.247 | 0.153 | 0.246 | 0.255 | 0.34 | 0.163 | 0.252 | 0.165 | 0.262 | 0.159 | 0.249 |
| | 192 | 0.171 | 0.260 | 0.169 | 0.260 | 0.264 | 0.349 | 0.175 | 0.263 | 0.184 | 0.278 | 0.171 | 0.260 |
| | 336 | 0.192 | 0.277 | 0.187 | 0.277 | 0.273 | 0.357 | 0.282 | 0.299 | 0.199 | 0.294 | 0.188 | 0.277 |
| | 720 | 0.220 | 0.304 | 0.227 | 0.311 | 0.31 | 0.384 | 0.357 | 0.348 | 0.231 | 0.318 | 0.228 | 0.309 |
| | AVG | 0.185 | 0.272 | 0.184 | 0.273 | 0.275 | 0.357 | 0.244 | 0.29 | 0.194 | 0.288 | 0.186 | 0.273 |
| Weather | 96 | 0.164 | 0.209 | 0.169 | 0.212 | 0.175 | 0.229 | 0.176 | 0.216 | 0.167 | 0.167 | 0.168 | 0.209 |
| | 192 | 0.222 | 0.261 | 0.218 | 0.256 | 0.223 | 0.264 | 0.223 | 0.255 | 0.217 | 0.256 | 0.213 | 0.250 |
| | 336 | 0.272 | 0.290 | 0.275 | 0.298 | 0.287 | 0.31 | 0.28 | 0.298 | 0.272 | 0.297 | 0.273 | 0.293 |
| | 720 | 0.359 | 0.352 | 0.353 | 0.349 | 0.364 | 0.358 | 0.357 | 0.349 | 0.353 | 0.349 | 0.354 | 0.347 |
| | AVG | 0.254 | 0.277 | 0.253 | 0.278 | 0.262 | 0.29 | 0.259 | 0.279 | 0.252 | 0.267 | 0.252 | 0.274 |
| Traffic | 96 | 0.439 | 0.300 | 0.447 | 0.303 | 0.648 | 0.405 | 0.442 | 0.302 | 0.440 | 0.309 | 0.433 | 0.290 |
| | 192 | 0.450 | 0.302 | 0.460 | 0.307 | 0.737 | 0.438 | 0.459 | 0.308 | 0.457 | 0.305 | 0.450 | 0.296 |
| | 336 | 0.462 | 0.310 | 0.482 | 0.319 | 0.745 | 0.435 | 0.479 | 0.319 | 0.478 | 0.321 | 0.467 | 0.306 |
| | 720 | 0.517 | 0.339 | 0.522 | 0.343 | 0.863 | 0.484 | 0.516 | 0.342 | 0.520 | 0.349 | 0.500 | 0.326 |
| | AVG | 0.467 | 0.312 | 0.477 | 0.318 | 0.748 | 0.44 | 0.474 | 0.317 | 0.473 | 0.321 | 0.462 | 0.304 |
| ETTh2 | 96 | 0.300 | 0.348 | 0.298 | 0.348 | 0.375 | 0.399 | 0.295 | 0.344 | 0.304 | 0.352 | 0.299 | 0.350 |
| | 192 | 0.389 | 0.402 | 0.387 | 0.402 | 0.45 | 0.435 | 0.375 | 0.398 | 0.391 | 0.406 | 0.384 | 0.401 |
| | 336 | 0.409 | 0.408 | 0.416 | 0.427 | 0.481 | 0.469 | 0.436 | 0.435 | 0.419 | 0.428 | 0.411 | 0.428 |
| | 720 | 0.421 | 0.439 | 0.426 | 0.443 | 0.471 | 0.474 | 0.429 | 0.447 | 0.422 | 0.438 | 0.430 | 0.446 |
| | AVG | 0.379 | 0.399 | 0.381 | 0.405 | 0.444 | 0.444 | 0.383 | 0.406 | 0.384 | 0.406 | 0.381 | 0.406 |
| Solar | 96 | 0.199 | 0.239 | 0.209 | 0.252 | 0.244 | 0.284 | 0.213 | 0.253 | 0.207 | 0.249 | 0.203 | 0.240 |
| | 192 | 0.233 | 0.263 | 0.238 | 0.274 | 0.277 | 0.306 | 0.242 | 0.274 | 0.240 | 0.279 | 0.237 | 0.268 |
| | 336 | 0.252 | 0.279 | 0.261 | 0.292 | 0.323 | 0.326 | 0.262 | 0.29 | 0.256 | 0.290 | 0.249 | 0.280 |
| | 720 | 0.259 | 0.280 | 0.265 | 0.295 | 0.311 | 0.329 | 0.27 | 0.296 | 0.275 | 0.303 | 0.254 | 0.283 |
| | AVG | 0.235 | 0.265 | 0.243 | 0.278 | 0.288 | 0.311 | 0.246 | 0.278 | 0.244 | 0.280 | 0.235 | 0.267 |
| PMES03 | 96 | 0.250 | 0.343 | 0.304 | 0.393 | 0.302 | 0.363 | 0.337 | 0.413 | 0.240 | 0.340 | 0.302 | 0.395 |
| | 192 | 0.297 | 0.378 | 0.353 | 0.424 | 0.373 | 0.422 | 0.374 | 0.436 | 0.300 | 0.384 | 0.349 | 0.428 |
| | 336 | 0.274 | 0.356 | 0.318 | 0.388 | 0.32 | 0.388 | 0.332 | 0.395 | 0.286 | 0.365 | 0.313 | 0.388 |
| | 720 | 0.336 | 0.401 | 0.387 | 0.438 | 0.409 | 0.44 | 0.397 | 0.44 | 0.352 | 0.414 | 0.383 | 0.441 |
| | AVG | 0.289 | 0.369 | 0.340 | 0.410 | 0.351 | 0.403 | 0.36 | 0.421 | 0.294 | 0.375 | 0.336 | 0.413 |
| PMES04 | 96 | 0.226 | 0.335 | 0.364 | 0.433 | 0.236 | 0.348 | 0.387 | 0.446 | 0.235 | 0.344 | 0.356 | 0.428 |
| | 192 | 0.286 | 0.383 | 0.406 | 0.406 | 0.326 | 0.413 | 0.429 | 0.473 | 0.307 | 0.395 | 0.398 | 0.459 |
| | 336 | 0.286 | 0.372 | 0.359 | 0.417 | 0.312 | 0.398 | 0.369 | 0.426 | 0.297 | 0.378 | 0.348 | 0.413 |
| | 720 | 0.338 | 0.414 | 0.436 | 0.471 | 0.387 | 0.451 | 0.44 | 0.476 | 0.362 | 0.428 | 0.423 | 0.467 |
| | AVG | 0.284 | 0.376 | 0.391 | 0.431 | 0.315 | 0.402 | 0.406 | 0.455 | 0.300 | 0.386 | 0.381 | 0.441 |
| PMES08 | 96 | 0.365 | 0.409 | 0.469 | 0.461 | 0.401 | 0.416 | 0.477 | 0.47 | 0.338 | 0.388 | 0.406 | 0.437 |
| | 192 | 0.513 | 0.462 | 0.612 | 0.507 | 0.567 | 0.481 | 0.627 | 0.518 | 0.487 | 0.444 | 0.539 | 0.479 |
| | 336 | 0.530 | 0.435 | 0.606 | 0.471 | 0.567 | 0.452 | 0.599 | 0.472 | 0.516 | 0.425 | 0.538 | 0.444 |
| | 720 | 0.624 | 0.487 | 0.692 | 0.531 | 0.697 | 0.512 | 0.691 | 0.532 | 0.600 | 0.479 | 0.603 | 0.489 |
| | AVG | 0.508 | 0.448 | 0.594 | 0.492 | 0.558 | 0.465 | 0.598 | 0.498 | 0.485 | 0.434 | 0.521 | 0.462 |

Table 15: Comparison of training time (s \iter) and GPU memory footprint (MB).

| Datasets Models | HTMFormer Ours | MPFormer 2025 | WPMixer 2025 | PatchTST 2023 |
|---|---|---|---|---|
| ECL | 0.1004 | 0.3319 | 0.4143 | 0.5088 |
| Weather | 0.0282 | 0.0320 | 0.0392 | 0.0292 |
| Traffic | 0.2682 | 0.9648 | 1.2374 | 1.0294 |
| ETTh2 | 0.0199 | 0.0279 | 0.0183 | 0.0190 |
| Solar | 0.0502 | 0.1435 | 0.1745 | 0.2200 |
| pems03 | 0.0982 | 0.3868 | 8344.82 | 0.5797 |
| pems04 | 0.1514 | 0.5801 | 0.3894 | 0.4857 |
| pems07 | 0.0915 | 0.3108 | 0.2154 | 0.2552 |
| AVG | 0.1010 | 0.3472 | 0.3849 | 0.3909 |

| Datasets Models | HTMFormer Ours | MPFormer 2025 | WPMixer 2025 | PatchTST 2023 |
|---|---|---|---|---|
| ECL | 1623.47 | 7471.94 | 13784.07 | 7195.99 |
| Weather | 151.83 | 584.33 | 651.45 | 538.76 |
| Traffic | 6221.31 | 20860.23 | 29560.07 | 19152.58 |
| ETTh2 | 102.48 | 265.89 | 455.10 | 430.54 |
| Solar | 647.04 | 3219.13 | 5970.26 | 3130.73 |
| pems03 | 1849.57 | 8344.82 | 8015.67 | 8015.67 |
| pems04 | 1536.60 | 7141.78 | 13195.91 | 6882.86 |
| pems07 | 778.26 | 3979.58 | 7369.67 | 3869.06 |
| AVG | 1613.82 | 6483.46 | 10125.27 | 6152.02 |

# H   HYPERPARAMETER SENSITIVITY

We assess the sensitivity of HTMformer to variations in key hyperparameters, including the learning rate $lr$, the number of Transformer blocks $L$, the FCN dimension $K$, and the hidden dimension $D$ of the variate tokens. This study can provide guidance for hyperparameter selection of HTMformer in practical applications. As shown in Figure 7, our analysis leads to the following key observations: Increasing hyperparameter values does not necessarily translate into improved forecasting accuracy. The model attains its best performance on most datasets when configured with $lr = 0.0005$, $L = 2$, $K = 1024$, and $D = 512$.

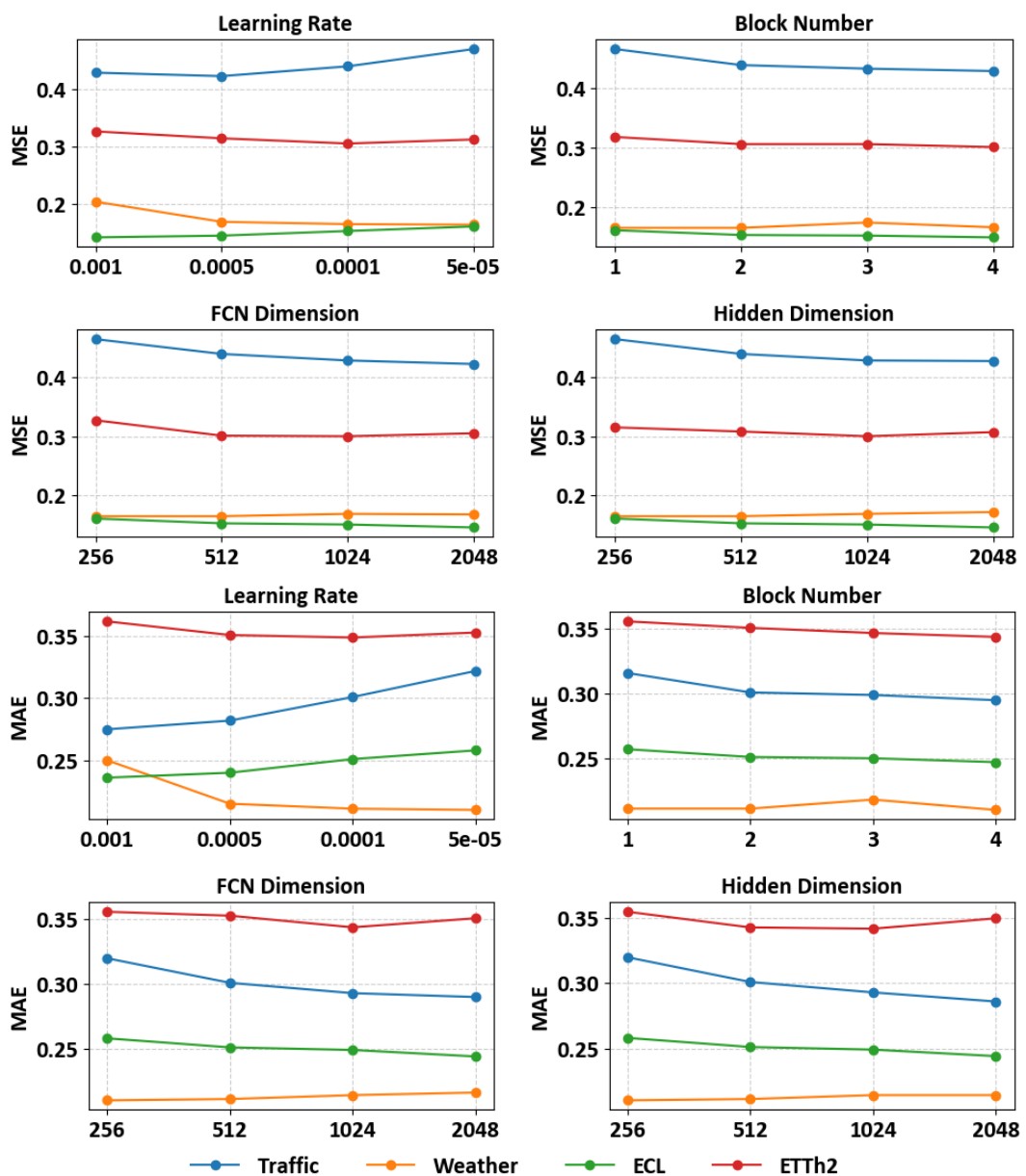

Figure 7: Hyperparameter sensitivity with respect to $lr$, $L$, $K$, and $D$ . All results are obtained using a lookback window of $T = 96$ time steps and a forecast horizon of $S = 96$ time steps.

## I Visualization of Prediction Results

We visualize HTMformer's ability to predict trends across various time series datasets, including ECL, Traffic, Solar-Energy, Weather, and PEMS (PEMS03, PEMS04 and PEMS08), as shown in Figures 8 to 15. Each example uses a 96-step input to generate 96-step predictions. In the visualizations, the orange lines indicate the ground truth values, and the blue lines show the model's predictions. It accurately captures the cyclical patterns and oscillatory behaviors, and successfully forecasts the overall directional trends.

To assess the performance of various models, we perform a qualitative comparison by visualizing the final forecasting results derived from two representative datasets (Weather and ETTh2). Among the various models, HTMformer exhibits superior or comparable performance in predicting the most series variations. Prediction showcases are listed in Figure 16 and Figure 17.

## J Discussion on Transformer-based Predictors

Recent studies (Lu et al., 2023; Li et al., 2023b) have found that Channel Dependence ideally gains from higher capacity, while Channel Independence can significantly enhance performance due to sample scarcity. In time series forecasting, an increasing number of models employ Channel Independence (Nie et al., 2023), as most current forecasting benchmarks are not sufficiently large. These models treat time series variates independently and utilize a shared backbone, achieving good performance. However, Channel Independence inevitably overlooks the multivariate correlations that exist in most datasets, resulting in the loss of some information. We believe that making variates independent is desirable, but multivariate correlations should not be ignored. However, previous improvements to models have mainly focused on extracting features in the temporal dimension. Excessive extraction of temporal features does not further improve prediction accuracy, but rather significantly increases computational overhead.

Notably, as demonstrated in both the introduction and ablation studies of this paper. The main reason why traditional Transformer-based predictors encounter performance bottlenecks is that the embedding model extracts only limited effective information from the sequence, which in turn affects the performance of subsequent models. Due to the inherent limitations of Transformer-based predictors, simply enlarging the input sequence length cannot further enhance accuracy. Constructing an embedding model that deeply extracts sequence information can effectively address this problem. Therefore, we propose a novel strategy, specifically extracting multivariate features in the embedding layer and using them to complement temporal features. This approach incorporates multivariate correlations, thereby further extracting effective information from the sequence. Moreover, it does not affect the subsequent attention mechanism to utilize Channel Independence.

We find that extracting local patterns in time series through patching and convolution, together with capturing long-term dependencies using linear layers, is sufficient for effectively modeling temporal features. For Transformer-based predictors, this is sufficient to extract temporal representations, and it is unnecessary to employ attention mechanisms to model temporal correlations.

Graph Convolutional Networks (GCNs) typically construct an adjacency matrix to represent multivariate correlations (Kipf & Welling, 2017; Bruna et al., 2014). However, the adjacency matrix is often non-learnable. Although the physical locations of sensors are relatively fixed, the correlations among variables are not necessarily constant, which severely affects the performance of GCNs. Introducing a learnable adjacency matrix (Wang et al., 2019) can substantially increase computational overhead and lead to model instability (Wu et al., 2020a; Franceschi et al., 2019). Therefore, this paper adopts a "weak learning" strategy, whereby multivariate correlations are simply extracted and incorporated as features into tokens to correct temporal features.

We validate the effectiveness and scalability of the proposed HTME strategy by integrating it into various Transformer variants. HTME can be seamlessly combined with various attention mechanisms across a wide spectrum of time series forecasting tasks to achieve optimal performance. Our model demonstrates superior performance in short-term sequence forecasting. On the selected datasets, PEMS03, PEMS04, and PEMS08, our model achieves the best results compared with the second-best methods. In long-term sequence forecasting, our model ranks first or second on eight selected datasets in most cases. This demonstrates the effectiveness of the HTME strategy in advancing time series

forecasting. We also compared the Inference Time, GPU Memory Footprint, and Model Parameters of HTMformer with those of three models that have good forecasting performance: MultiPatchFormer, WPMixer, and PatchTST. On the selected datasets, HTMformer achieves superior or equivalent performance with only about 0.2-0.45 times the memory of the three models, and is two to three times faster. These results suggest that the HTME model is highly flexible for deployment across a broader spectrum of devices and can better accommodate real-time forecasting tasks, thereby creating greater potential for practical applications and future scalability.

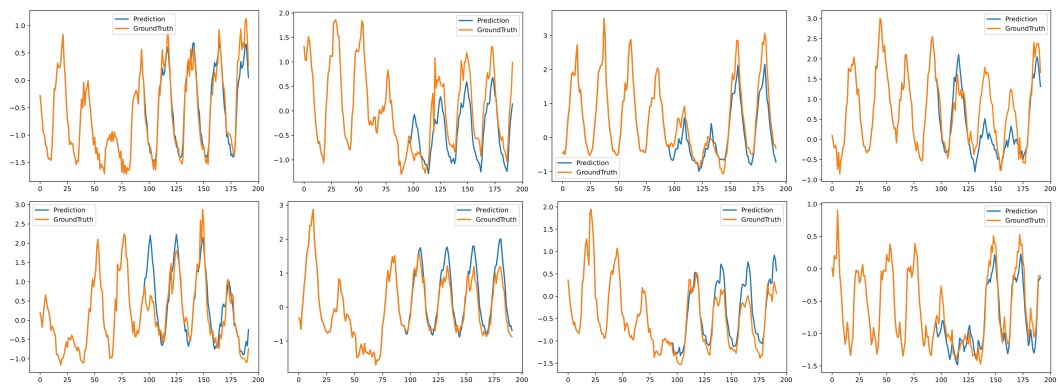

Figure 8: Examples of forecasts for the ECL dataset with a 96-step predictions.

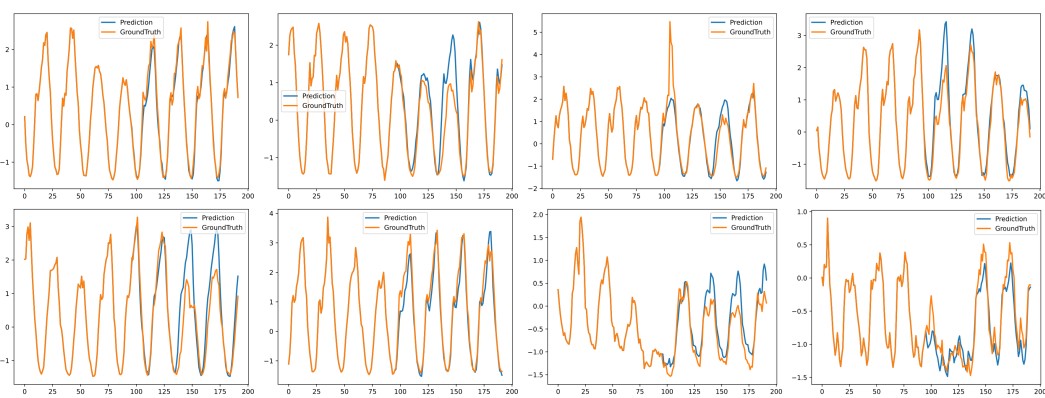

Figure 9: Examples of forecasts for the Traffic dataset with a 96-step predictions.

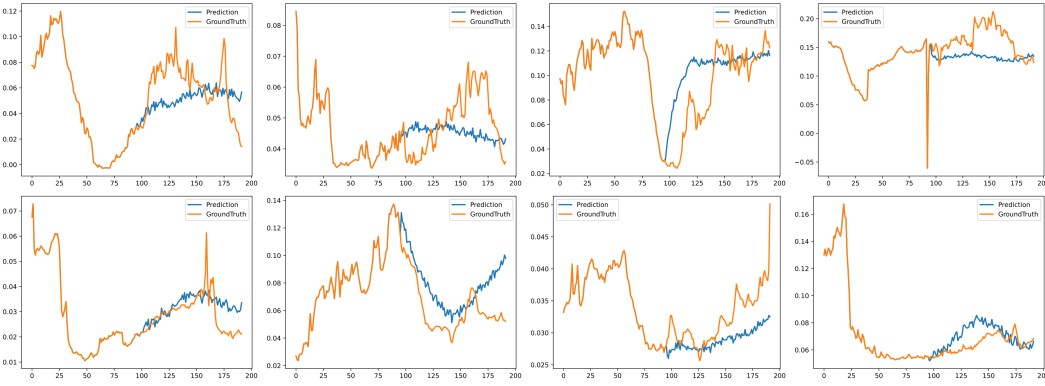

Figure 10: Examples of forecasts for the Weather dataset with a 96-step predictions.

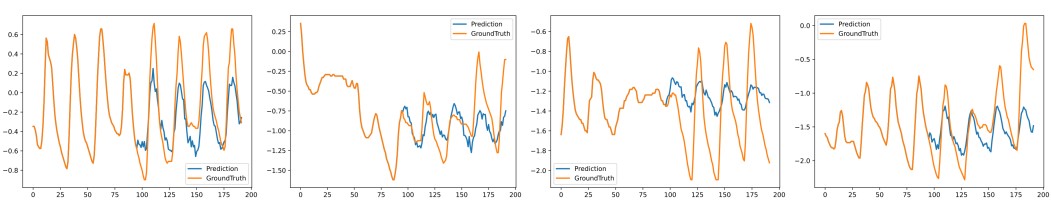

Figure 11: Examples of forecasts for the ETTh2 dataset with a 96-step predictions.

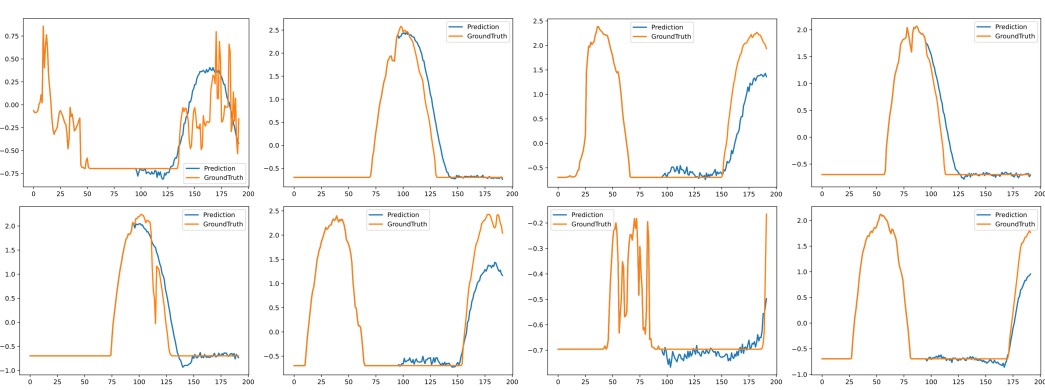

Figure 12: Examples of forecasts for the Solar dataset with a 96-step predictions.

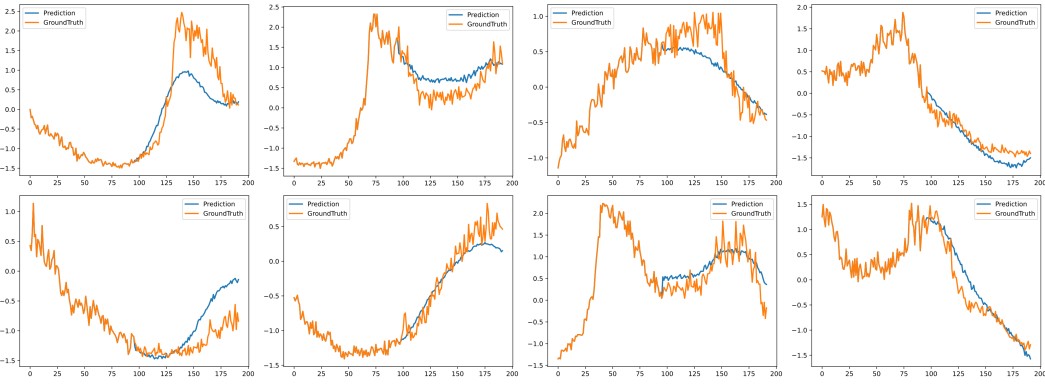

Figure 13: Examples of forecasts for the PEMS03 dataset with a 96-step predictions.

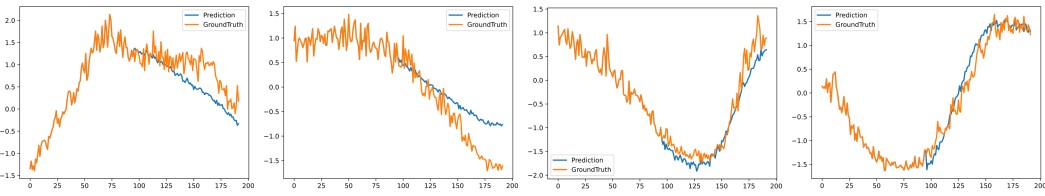

Figure 14: Examples of forecasts for the PEMS04 dataset with a 96-step predictions.

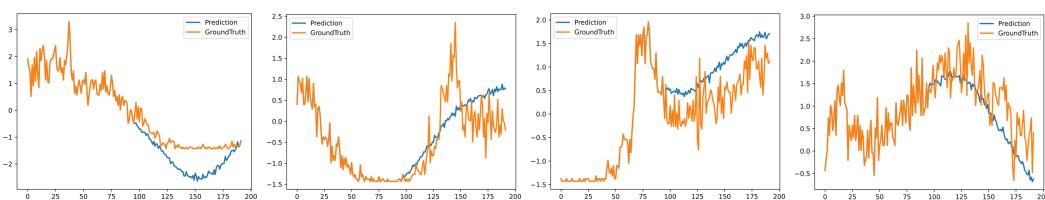

Figure 15: Examples of forecasts for the PEMS08 dataset with a 96-step predictions.

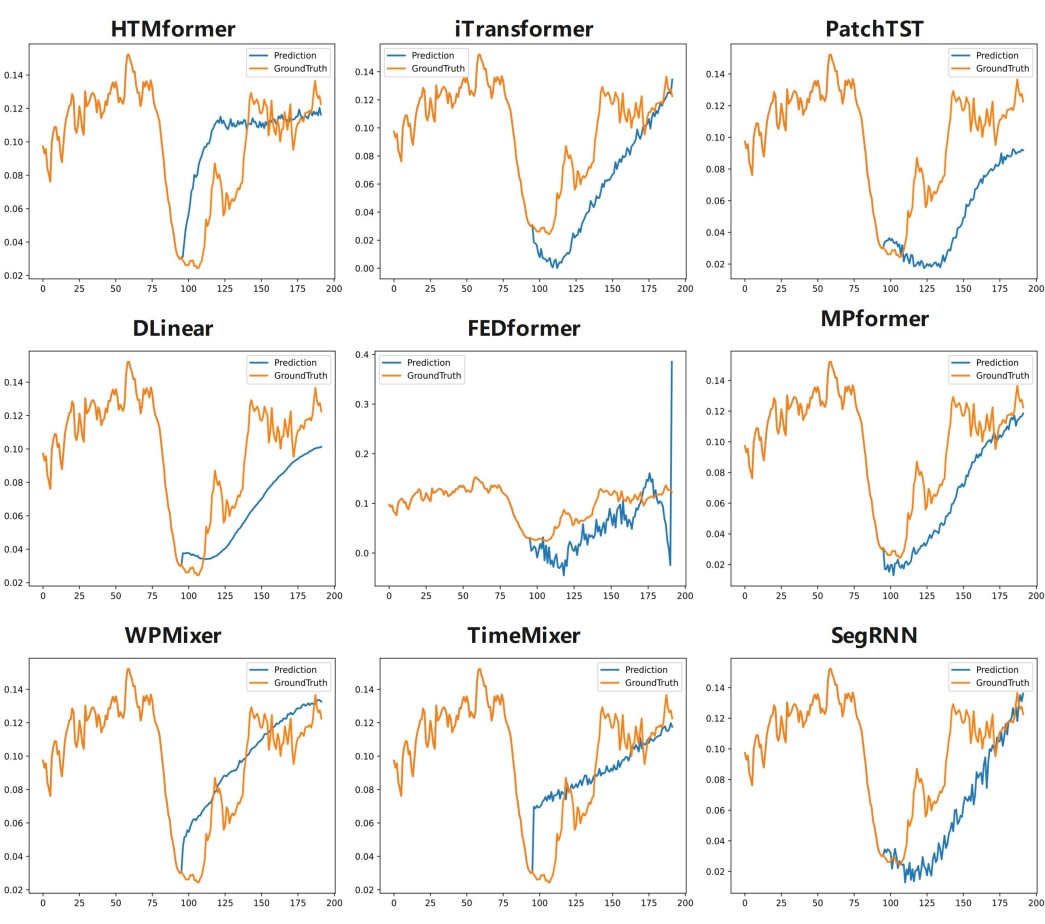

Figure 16: Visualization of results on the Weather dataset across all selected models.

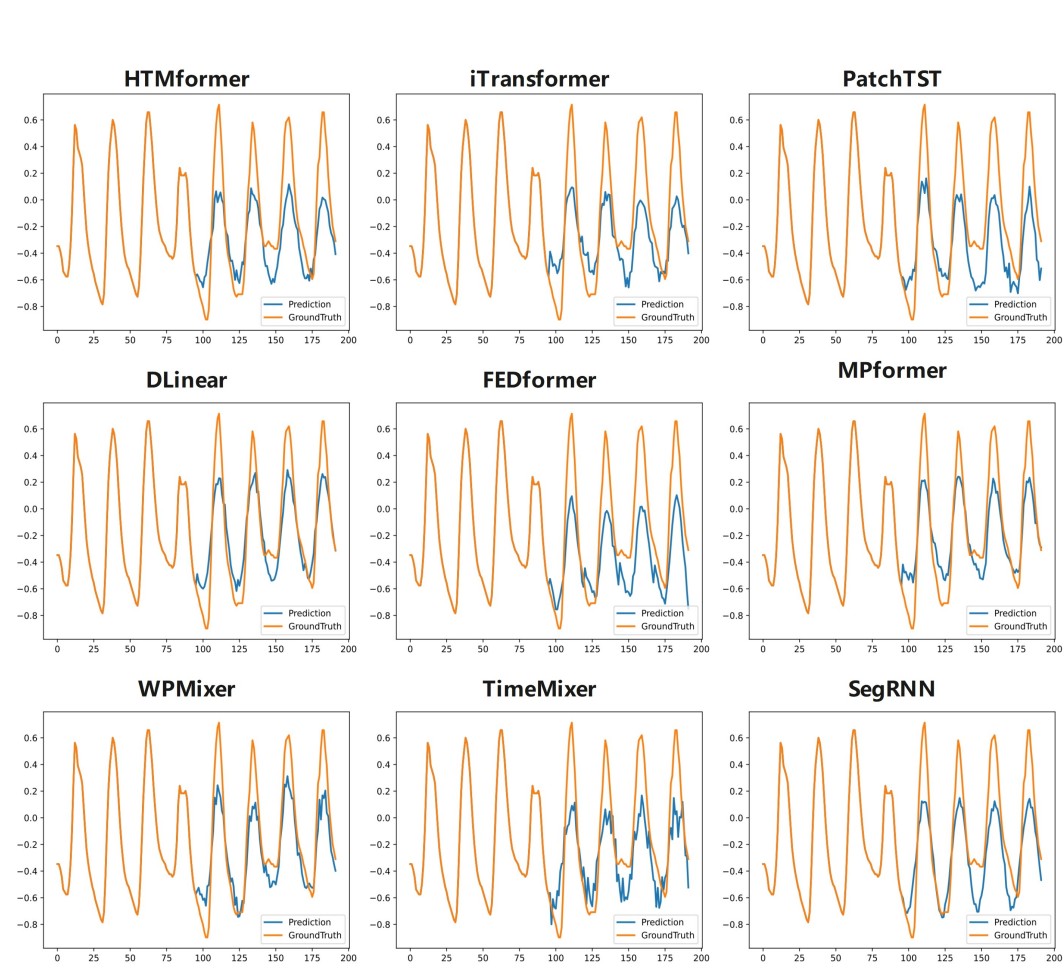

Figure 17: Visualization of results on the ETTh2 dataset across all selected models.

