# OpenReview forum: "HTMformer: Hybrid Time and Multivariate Transformer for Time Series Forecasting"
_ICLR.cc/2026/Conference — ICLR 2026 Conference Withdrawn Submission_

### Official Review · Reviewer_Hy2y · 2025-10-25

**Soundness:** 2
**Presentation:** 2
**Contribution:** 2
**Rating:** 2
**Confidence:** 5

**Summary:**

This paper introduces HTMformer, a model that integrates a custom-designed Hybrid Temporal and Multivariate Embedding (HTME) module. HTME integrates a temporal feature extraction module with a multivariate feature extraction module to provide complementary features.

**Strengths:**

1. The method is very simple but effective. Different from works focusing on model architecture design, this paper focuses on the embedding modeling.

**Weaknesses:**

1. The integration of temporal and multivariate features represents a well-trodden path in multivariate time series forecasting. The work presented in this paper does not offer distinctly new insights or conceptual advancements beyond this established paradigm, resulting in a lack of compelling novelty.

2. The HTME module is positioned as the core contribution of this work. However, its architectural design appears to be a straightforward composition of existing feature extraction techniques. The paper would benefit from a clearer elaboration of the core design principle or the key innovation that distinguishes HTME from a mere combination of established components.

3. In Table 1, why not apply HTME to models like PatchTST to observe its performance improvement? Baselines like Informer and Reformer are too old.

4. A high-quality embedding strategy should possess general applicability across different model architectures. Is the proposed HTME compatible with non-Transformer models, such as DLinear?

**Questions:**

See weaknesses.

---

> ### Author Response · Authors · 2025-11-19
>
> 1. Thank you for raising this important point. We agree that combining temporal and multivariate features is a common direction in multivariate time series forecasting. However, our work does not simply follow this established paradigm.
> We point out that existing predictors for multivariate correlation modeling tend to adopt an explicit modeling approach, which is computationally expensive and confounds temporal and multivariate features.Our approach of supplementing temporal features with multivariate features significantly simplifies the multivariate correlation modeling process and ensures computational efficiency.
>
>    The core contribution lies in proposing a new way to operationalize this combination through a parallel extraction and fusion framework together with a novel strategy for hierarchical temporal modeling. Directly extracting features in parallel from the raw sequences minimizes noise from other dimensions. As a preprocessing module, HTME exhibits strong compatibility. As demonstrated in our experiments, integrating our hybrid embeddings into various existing backbones consistently improves performance while lowering computational cost. This indicates that the contribution is not merely another feature combination, but rather a new embedding paradigm that enhances the fundamental representational ability of time series models. Taken together, these elements provide conceptual and methodological innovations that meaningfully extend beyond the existing paradigm of integrating temporal and multivariate features.
>
> 2. Thank you for the valuable comment. The innovation lies in a design philosophy for overcoming the representational bottlenecks of Transformer-based forecasting models..HTME instead isolates the two forms of information and models them with lightweight but specialized extractors before fusing them into a unified representation. This structural disentanglement allows each extractor to focus on a distinct aspect of the data rather than sharing limited embedding capacity. The second principle is the aggregate then decompose strategy used for hierarchical temporal modeling. Instead of trying to learn long-term dynamics directly in one step, HTME first aggregates short segments to capture essential short-term statistics and then gradually decomposes the collected information to reveal more global temporal dependencies. This approach encourages the model to concentrate on the most informative short-term patterns while still retaining long-term awareness, and it effectively reduces noise and computational cost. These design principles together give HTME a coherent and targeted structure that improves the expressive power of the embeddings in a way that established components alone cannot achieve. The experimental results show that this paradigm enhances a wide range of backbone models, confirming that HTME is not a simple combination of existing techniques but a principled embedding framework that strengthens time series representations.
>
>    We detail the mechanisms and mathematical principles of HTME in Section 3.
>
> 3. 1.Thank you for the question. The improvements of PatchTST also lie in the embedding layer, leaving other modules unchanged. Consequently, PatchTST and HTME are architecturally incompatible. Applying HTME requires modifying its core patching design rather than simply attaching an additional embedding module. For this reason, the models in Table 1 focus on architectures with simple and compatible embedding layers, such as Reformer, Informer, Flowformer and Flashformer, so that the contribution of HTME can be evaluated without altering their fundamental structures.
>
> 4. Yes, HTME is compatible with non Transformer models. HTME acts as a data enhancement oriented preprocessing layer, so in principle it can be integrated with any forecasting architecture. To verify its general applicability, we incorporate HTME into DLinear as a representative linear model. The experimental results show consistent performance improvements in both settings, demonstrating that the effectiveness of HTME is not limited to Transformer based predictors. This set of experiments also aims to address Question 3
>
> I am finalizing the revisions and will send the updated PDF to you very soon. I would greatly appreciate any further suggestions you may have.

---

> ### Author Response · Authors · 2025-11-23
>
> Following your valuable suggestions, we have revised the manuscript accordingly. The specific modifications are as follows:
>
> 1. In the Introduction and Conclusion sections, we have amplified our discussion on the limitations of existing models and the contributions of HTME. Specifically, we now emphasize that current models often fail to strike an effective balance between computational cost and feature extraction. We address this challenge through a simple yet effective preprocessing layer.
> 2. We have elaborated on the principles of HTME in Section 3, while also removing redundant descriptions to improve clarity.
> 3. In Section 4, we have added experiments that combine HTME with other backbone architectures. The results demonstrate that HTME consistently enhances the representation learning capabilities of various models.
> 4. A complexity analysis of HTME has been added to Section 3.4.
> 5. We made minor revisions to correct grammatical errors and improve readability.

---

### Official Review · Reviewer_A3yp · 2025-10-31

**Soundness:** 3
**Presentation:** 2
**Contribution:** 2
**Rating:** 2
**Confidence:** 3

**Summary:**

The manuscript proposes HTME, a feature-extraction and embedding-generation module, and builds HTMformer on top of it to model dependencies across variable dimensions of multivariate time-series embeddings using a Transformer encoder.

**Strengths:**

The proposed HTMformer is evaluated on standard benchmark datasets and reports promising accuracy compared with recent state-of-the-art baselines.

The HTME extractor is designed to capture local temporal dependencies and produce a fused representation (temporal + spatial) that serves as input to the Transformer encoder.

The overall architecture is encoder-only and relatively lightweight, which is attractive for long-horizon forecasting settings.

**Weaknesses:**

The description of HTME is not sufficiently detailed. A step-by-step explanation of the computations is needed to make the core contribution reproducible.

The rationale for fusing temporal and spatial representations before the Transformer is unclear. It is not evident why summation is the best fusion operator, nor how it affects the disentangling of temporal vs. variable-wise patterns.

The use of GRU inside HTME is under-motivated, especially given that the main model is Transformer-based; it is not explained why a Transformer (or even a lightweight temporal attention) is not used in this stage.

The current HTME illustration (Figure 3) does not fully illustrate the computation described in the text and should be revised.

**Questions:**

(a) HTMformer is missing from Figure 1.

(b) Provide explicit dimensions for all components in Equations (2), (3), and (4). This is the core part of the contribution. Also, clarify why an additional linear layer is used to project from $dim$ to $D$ in Eq. (4) instead of directly projecting the flattened vectors to $D$.

(c) Explain the motivation for summing $D_{\text{out}}$ and $V_{\text{out}}$. Does this operation beneficially entangle temporal and spatial features, or would a decoupled design (two Transformers, one temporal and one variable-wise, followed by fusion) be more appropriate?

(d) Please add an ablation that replaces the GRU in Eq. (6) with a Transformer-based alternative to verify that GRU is indeed the better choice.

(e) Section 3.3 can be made more concise, since it mainly reuses existing models and may distract from the contributions (HTME).

(f) The paper states that HTMformer changes the complexity from $O(L^2)$ to $O(N^2)$. Please discuss the case where $N$ is greater than $L$ and whether the proposed formulation still offers computational advantages.

(g) In Figure 4, replace training time with FLOPs to make comparisons hardware-agnostic.

---

> ### Author Response · Authors · 2025-11-19
>
> A: Thank you for pointing this out. This was a labeling mistake. The red-highlighted model in Figure 1 is our HTMFormer, and the figure has now been corrected to accurately reflect our model.
>
> B,C,D,E：In accordance with your suggestions, Section 3 has been rewritten. We now begin the section with a more detailed explanation of the HTME mechanism and its core motivation. Subsequently, the remainder of the section has been streamlined to eliminate redundancy and enhance clarity.
>
> C:Why do we use addition instead of other feature fusion methods? The motivation for addition is to efficiently integrate temporal and multivariate features while preserving our design goal that multivariate features complement rather than dominate temporal features. In the Introduction, we emphasize that this fusion aims to produce embeddings with richer semantics. Prior methods, such as GNNs or Crossformer, explicitly model multivariate correlations in the backbone, but they incur substantial computational cost. Alternative fusion strategies, including separate Transformers for each dimension followed by concatenation or other parallel structures, would also increase computation and reduce scalability without significant performance gains. Addition provides a simple, lightweight, and generalizable approach that allows HTME to be attached to nearly any architecture while effectively combining the two types of features.
>
> D:I will proceed with the experiments as you have requested. I will get back to you once they are complete.
>
>
>
> F: We added supplementary explanations on computational complexity, mainly covering two aspects:
>   1. iTransformer reports that using attention to capture multivariate correlations is more effective than focusing on temporal correlations. Therefore, we adopt the inverted input structure (see Section 3.3 — Encoder and Projection), and the resulting reduction in complexity is a potential benefit.
>   2. The benefit is clearer from the properties of real datasets: experiments in Appendix A show that the optimal look-back window length is typically around 96–144, while the number of variables rarely exceeds 100. Therefore, in most cases, O(N^2) < O(L^2).
> Even when O(N^2) > O(L^2), our model still achieves significant advantages in comparisons due to our effective module design. The experimental results confirm that our model does not only perform well on high-dimensional datasets. Additionally, since N is intrinsic to the dataset, we can adjust the look-back window length according to the dataset without worrying about additional computational burden.
>
> It is also important to note that the computational cost comparisons were conducted using exactly the same machine and configuration to ensure fairness.
>
> G: This experiment will take some time to complete. I will report the results to you as soon as it is finished.

---

> ### Author Response · Authors · 2025-11-19
>
> Thank you for your valuable feedback. All the suggested revisions have been incorporated into the manuscript. I will now submit the revised PDF and provide a point-by-point response detailing the location of each modification.I am happy to clarify any points or answer any questions you might have. Any suggestions you could offer to help improve this work would be greatly appreciated.

---

> ### Author Response · Authors · 2025-11-23
>
> Following your valuable suggestions, we have revised the manuscript accordingly. The specific modifications are as follows:
> 1. In the Introduction and Conclusion sections, we have amplified our discussion on the limitations of existing models and the contributions of HTME. Specifically, we now emphasize that current models often fail to strike an effective balance between computational cost and feature extraction. We address this challenge through a simple yet effective preprocessing layer.
> 2. We have elaborated on the principles of HTME in Section 3, while also removing redundant descriptions to improve clarity.
> 3. In Section 4, we have added experiments that combine HTME with other backbone architectures. The results demonstrate that HTME consistently enhances the representation learning capabilities of various models.
> 4. A complexity analysis of HTME has been added to Section 3.4.
> 5. We made minor revisions to correct grammatical errors and improve readability.

---

> ### Author Response · Authors · 2025-11-23
>
> D: As requested, we replaced the GRU module with the Transformer module. However, this change led to a decline in model performance. We attribute this to the Transformer's tendency to over-model multivariate correlations. Furthermore, the complex architecture of the Transformer also resulted in a significant increase in computational overhead.
> | Dataset | Length | Former (MSE) | Former (MAE) | GRU (MSE) | GRU (MAE) |
> |:--------|:-------|:-------------|:-------------|:----------|:----------|
> | ELC     | 96     | 0.168        | 0.255        | 0.157     | 0.247     |
> | ELC     | 192    | 0.18         | 0.265        | 0.171     | 0.26      |
> | ELC     | 336    | 0.199        | 0.285        | 0.192     | 0.277     |
> | ELC     | 720    | 0.258        | 0.331        | 0.22      | 0.304     |
> | ELC     | **AVG**  | **0.201**      | **0.284**      | **0.185**   | **0.272**   |
> | WEATHER | 96     | 0.211        | 0.262        | 0.167     | 0.211     |
> | WEATHER | 192    | 0.246        | 0.281        | 0.222     | 0.261     |
> | WEATHER | 336    | 0.309        | 0.328        | 0.272     | 0.290     |
> | WEATHER | 720    | 0.371        | 0.362        | 0.359     | 0.352     |
> | WEATHER | **AVG**  | **0.284**      | **0.308**      | **0.255**   | **0.278**   |
>
> | Dataset     | Former  | Gru     |
> |-------------|---------|---------|
> | Weather     |  4.277G  | 3.803G  |
> | Electricity |  58.122G | 50.100G |
>
> G: As requested, we have added supplementary experiments. We compared the floating-point operations (FLOPs) under a unified experimental setup. The look-back window size was set to 96, with a prediction horizon of 192.
> | Dataset     | HTMformer | PatchTST  | MPformer | WPMixer   |
> |-------------|-----------|-----------|----------|-----------|
> | Weather     | 3.803G    | 34.764G   | 94.122G  | 43.261G   |
> | Electricity | 50.100G   | 531.396G  | 1.439T   | 661.165G  |

---

### Official Review · Reviewer_v2C6 · 2025-10-31

**Soundness:** 3
**Presentation:** 2
**Contribution:** 2
**Rating:** 4
**Confidence:** 3

**Summary:**

The paper proposes HTMformer, a novel time series forecasting framework using a Hybrid Temporal and Multivariate Embedding (HTME) module. HTME disentangles temporal and multivariate features by separately extracting (1) temporal patterns via patch-based convolution and linear projection, and (2) multivariate dependencies through a GRU-based pipeline (patch->linear->GRU->convolution), which are then fused with learnable weights. Integrated with an "inverted input" Transformer encoder, HTMformer achieves state-of-the-art performance on 8 benchmark datasets (e.g., Traffic, Solar-Energy) for both short- and long-term forecasting, while reducing training time and GPU memory usage. Ablation studies confirm the superiority of the hybrid design over single-feature baselines, and the method generalizes well across Transformer variants.

**Strengths:**

1.	The proposed HTME module separates temporal and multivariate feature extraction while maintaining their complementarity. This design not only draws on PatchTST's focus on local temporal patterns but also reflect iTransformer's emphasis on the relationships between variables.
2.	Experiments across eight widely used benchmarks (Weather, Traffic, Electricity, ETTh2, Solar, and multiple PEMS datasets), including comparisons, ablation studies, and evaluations of HTME integrated into various Transformer variants. This demonstrates the method’s stability and scalability, consistently outperforming baseline models.
3.	HTMformer achieves 2~3× faster training speed and uses only 20%~45% of the GPU memory compared to other methods.

**Weaknesses:**

1.	The proposed HTMformer is built on the inverted architecture of iTransformer, and does not propose a new backbone structure or attention mechanism, which lacks paradigm-level innovation.
2.	While the paper empirically shows that the multivariate-only variant (HTMformerV2) performs poorly on temporally dominant datasets (e.g., Traffic, ETTh2) but better on highly correlated ones (e.g., PEMS), it provides no causal explanation for when and why each module is effective. The authors do not link dataset characteristics (e.g., variable number, forecastability, periodicity) to module performance.
3.	The author overemphasizes the role of multivariate and weakens the foundation of time modeling. However, experiments (Table 4) show that time dependence is still the main reason for performance, but the paper narrative can be misleading.
4.	There are no comparative experiments to show why early decoupling time and multivariate is a better inductive bias. The author also did not point out the rationality of early decoupling from the perspective of decoupling theory or information bottlenecks.

**Questions:**

Could the authors provide a more principled or analytical justification for the HTME design? For example, could a simple mathematical formulation (e.g., between the embedding structures of HTME and iTransformer’s MLP embedding) help illustrate how HTME better preserves cross-variable temporal dynamics? Offering even a high-level analytical perspective would clarify the conceptual motivation behind HTME and help distinguish it from prior component-wise engineering improvements.

---

> ### Author Response · Authors · 2025-11-19
>
> 1. Thank you for the insightful comment. While HTMformer is indeed built upon the inverted architecture of iTransformer, our contribution goes beyond proposing a new backbone or attention mechanism. The key paradigm-level innovation of our work lies in shifting the focus from architectural changes to enhancing the representational capacity of input embeddings, which we find to be the true bottleneck of Transformer-based time series forecasting. Specifically, we introduce a hybrid temporal and multivariate embedding method that explicitly extracts lightweight temporal statistical features together with high-order multivariate interaction features. This parallel extraction and fusion paradigm provides richer and more meaningful sequence representations, which can be seamlessly integrated into various architectures. Additionally, our aggregate then decompose strategy models short-term correlations prior to long-term correlations, allowing the model to focus on the most essential temporal patterns while reducing noise and computational cost. As demonstrated in our experiments, this design consistently improves performance across diverse backbone architectures and maintains strong accuracy with significantly lower computational overhead.
> Our work emphasizes two key points:
>    1.  The enhanced embedding representation fully accounts for multivariate correlations, endowing HTME with richer feature information. This allows it to consistently enhance the modeling capabilities of various architectures, thereby breaking through a critical bottleneck faced by traditional time-series forecasting models.
>    2. HTME possesses strong generalizability. It functions merely as a data preprocessing module and can be integrated with any prediction architecture to consistently improve its performance.
>
>    Thus, the core innovation of HTMformer lies in establishing a new embedding paradigm that substantially enhances the effectiveness of existing time series forecasting models.
> 2. We provide a detailed explanation of the HTME mechanism, including its working principles and formula derivations.
> 3. As noted in the Introduction, while temporal information is mainly stored along the time dimension, the multivariate dimension also contains complementary information that can enrich the representation. Our core contribution lies in explicitly leveraging these multivariate features without weakening the modeling of temporal dependencies. We propose two modules: one for temporal feature extraction and another for multivariate feature extraction (see Section 3.1 — HTMEExtractor). To clarify their respective contributions, we conducted ablation studies. HTMFormerV1 uses only the temporal feature module, and HTMFormerV2 uses only the multivariate feature module. To further validate the benefit of multivariate information, we incorporated the multivariate extraction module into iTransformer to form iTransV3. The results demonstrate that the temporal feature extraction module remains the primary driver of performance, while the multivariate module consistently provides complementary gains across different temporal extractors (see Section 5.1 — Ablation Studies).
> 4. We acknowledge the need to justify the early decoupling of temporal and multivariate feature extraction. Many existing models, such as the FCN layers in Transformers, implicitly capture multivariate features, but they often prioritize temporal modeling.we provide a theoretical rationale for early decoupling: separating temporal and multivariate extraction at the embedding stage allows each module to independently capture information along its respective dimension, reducing interference and improving representation quality. From the perspective of decoupling theory and information bottlenecks, early separation ensures that temporal and multivariate information is processed without unnecessary compression or entanglement, which can otherwise obscure relevant patterns. Moreover, embedding-level separation enables a parallel structure, making HTME highly generalizable and easily deployable in front of various predictors. Table 1 empirically demonstrates that this approach consistently enhances the performance of diverse models, supporting the effectiveness of early decoupling.
>
> Should you have any further inquiries, please do not hesitate to contact me. I would be most grateful for your valuable comments and suggestions.

---

> ### Author Response · Authors · 2025-11-23
>
> Following your valuable suggestions, we have revised the manuscript accordingly. The specific modifications are as follows:
> 1. In the Introduction and Conclusion sections, we have amplified our discussion on the limitations of existing models and the contributions of HTME. Specifically, we now emphasize that current models often fail to strike an effective balance between computational cost and feature extraction. We address this challenge through a simple yet effective preprocessing layer.
> 2. We have elaborated on the principles of HTME in Section 3, while also removing redundant descriptions to improve clarity.
> 3. In Section 4, we have added experiments that combine HTME with other backbone architectures. The results demonstrate that HTME consistently enhances the representation learning capabilities of various models.
> 4. A complexity analysis of HTME has been added to Section 3.4.
> 5. We made minor revisions to correct grammatical errors and improve readability.

---

> > ### Comment · Reviewer_v2C6 · 2025-11-26
> > **Thank the authors for their responses**
> >
> > Thank the authors for their responses, but several significant flaws remain in the revised manuscript. First, the revised Figure 3 introduces severe errors. The input label is misspelled as "Time Serise Input" instead of "Series", and the operation arrow is labeled "Stock" instead of "Stack."  This contradicts the paper's text and Equation (2).Additionally, while the authors have added Equations (1)–(8) in Section 3.1, these are purely descriptive translations of the code logic and lack analytical depth.  The analysis in Section 5.1 remains qualitative, and table 14 shows that HTMformerV1 outperforms the HTMformer on certain datasets (e.g., ECL, Weather). The authors have not quantitatively linked dataset characteristics (such as the 'Forecastability' scores in Table 7) to the specific performance gains of the temporal vs. multivariate modules, as requested for a causal explanation. Therefore, I choose to maintain the original score.

---

### Comment · Area_Chair_Kqut · 2025-11-25

Dear Reviewers,

Thank you for reviewing for ICLR. Since the discussion deadline is coming soon, could you please take a look at the author's rebuttal, respond to their comments, and update your rating as well? Thanks!

Best Regards
AC

---

### Note · Authors · 2025-12-01

**Comment:**

I am planning to withdraw and revise the article.

**Withdrawal Confirmation:**

I have read and agree with the venue's withdrawal policy on behalf of myself and my co-authors.